# Loss of endothelial glucocorticoid receptor accelerates diabetic nephropathy

Swayam Prakash Srivastava[1,2,3], Han Zhou[1,2], Ocean Setia[2,4], Bing Liu[1,2], Keizo Kanasaki [3], Daisuke Koya [3], Alan Dardik [2,4,5], Carlos Fernandez-Hernando [2,6,7,8] & Julie Goodwin [1,2 ✉]

Endothelial cells play a key role in the regulation of disease. Defective regulation of endothelial cell homeostasis may cause mesenchymal activation of other endothelial cells or neighboring cell types, and in both cases contributes to organ fibrosis. Regulatory control of endothelial cell homeostasis is not well studied. Diabetes accelerates renal fibrosis in mice lacking the endothelial glucocorticoid receptor (GR), compared to control mice. Hypercholesterolemia further enhances severe renal fibrosis. The fibrogenic phenotype in the kidneys of diabetic mice lacking endothelial GR is associated with aberrant cytokine and chemokine reprogramming, augmented Wnt signaling and suppression of fatty acid oxidation. Both neutralization of IL-6 and Wnt inhibition improve kidney fibrosis by mitigating mesenchymal transition. Conditioned media from endothelial cells from diabetic mice lacking endothelial GR stimulate Wnt signaling-dependent epithelial-to-mesenchymal transition in tubular epithelial cells from diabetic controls. These data demonstrate that endothelial GR is an essential antifibrotic molecule in diabetes.

[1] Department of Pediatrics, Yale University School of Medicine New Haven, New Haven, CT, USA. [2] Vascular Biology and Therapeutics Program, Yale University School of Medicine New Haven, New Haven, CT, USA. [3] Department of Diabetology and Endocrinology, Kanazawa Medical University, Uchinada, Japan. [4] Department of Surgery, Yale University School of Medicine New Haven, New Haven, CT, USA. [5] Department of Surgery, VA Connecticut Healthcare System, West Haven, CT, USA. [6] Department of Comparative Medicine, Yale University School of Medicine New Haven, New Haven, CT, USA. [7] Program in Integrative Cell Signaling and Neurobiology of Metabolism (ICSNM), Yale University School of Medicine New Haven, New Haven, CT, USA. [8] Department of Pathology, Yale University School of Medicine New Haven, New Haven, CT, USA. ✉email: Julie.goodwin@yale.edu

Approximately one-third of diabetic patients will develop diabetic nephropathy (DN), a leading cause of end-stage renal disease that causes more than 950,000 deaths globally each year[1,2]. Over the last two decades, no new drugs have been approved to specifically prevent DN or to improve kidney function[3]. This lack of advancement stems, in part, from poor understanding of the mechanisms of progressive kidney dysfunction. Furthermore, this knowledge gap contributes to suboptimal treatment options available for these patients. Improved understanding of mechanisms of pathogenesis of diabetic kidney disease is urgently needed to catalyze the development of novel, effective, and safe therapeutics, which can be targeted to the early stages of diabetes, before kidney damage becomes irreversible.

DN is characterized by excess deposition of extracellular matrix, loss of capillary networks, and accumulation of fibrillary collagens and activated myofibroblasts and inflammatory cells[4,5]. In renal fibrosis, myofibroblasts are believed to be an activated fibroblast phenotype that contributes to fibrosis[5,6]. There are six well-reported sources of matrix-producing myofibroblasts: (1) activated residential fibroblasts, (2) differentiated pericytes, (3) recruited circulating fibrocytes, (4) those from macrophages via macrophage-to-mesenchymal transition, (5) those from mesenchymal cells derived from tubular epithelial cells (TECs) via epithelial-to-mesenchymal transition (EMT), and (6) those from mesenchymal cells transformed from endothelial cells (ECs) via endothelial-to-mesenchymal transition (EndMT)[7,8].

Among these diverse sources of matrix-producing fibroblasts, mesenchymal cells transformed from ECs via EndMT[7,8], are an important source of myofibroblasts in several organs, including the kidney[9]. EndMT is characterized by the loss of endothelial markers, including cluster of differentiation 31 (CD31), and acquisition of the expression of mesenchymal proteins, including α-smooth muscle actin (αSMA), vimentin, and fibronectin[7,8].

ECs are critical contributors to the formation of new blood vessels in health and life-threatening diseases[10]. Disruption in the central metabolism of ECs contributes to disease phenotypes[11,12]. Carnitine palmitoyltransferase 1a (CPT1a)-mediated fatty acid oxidation (FAO) regulates the proliferation of ECs in the stalk of sprouting vessels[13–15]. ECs use metabolites/precursors for epigenetic regulation of their subtype differentiation and maintain crosstalk through metabolites released by other cell types[10,15]. Notably, EndMT causes alteration of EC metabolism, and is an area of active investigation[16,17]. For example, mesenchymal cells derived from EndMT reprogram their metabolism and show defective fatty acid (FA) metabolism[17].

The contribution of EndMT to renal fibrosis has been analyzed in several mouse models of chronic kidney disease[5–7,18,19]. Zeisberg et al. performed seminal experiments and reported that ~30–50% of fibroblasts co-expressed the EC marker CD31 along with markers of fibroblasts and myofibroblasts such as fibroblast-specific protein-1 and αSMA in the kidneys of mice subjected to unilateral ureteral obstruction nephropathy (UUO)[19]. The complete conversion from EC into mesenchymal cell types is not needed as intermediate cell types are sufficient to cause the activation of profibrogenic pathways. EndMT can induce profibrogenic signaling in neighboring cells by autocrine and/or paracrine mechanisms thereby contributing to global kidney fibrosis[6,20,21]. Thus, targeting EndMT might have therapeutic potential for the treatment of renal fibrosis[6,19,22].

The glucocorticoid receptor (GR) is a nuclear hormone receptor that is expressed ubiquitously in most cell types and is important in many states of health and disease. GRs mediate the action of steroid hormones in a variety of tissues, including the kidney. Our previous work has demonstrated that tissue-specific loss of this receptor can produce profound phenotypes[23–26]. The role of glucocorticoids in cardiovascular and kidney disease is complex. We have identified endothelial GR as a negative regulator of vascular inflammation in models of sepsis[23] and atherosclerosis[25]. Recently, we demonstrated loss of endothelial GR results in upregulation of the canonical Wnt signaling pathway[27]. Notably, this pathway is known to be upregulated in renal fibrosis[28]. However, whether endothelial GR contributes to the regulation of fibrogenic processes in the evolution of kidney fibrosis is not known.

In this work, we show that mice lacking endothelial GR display accelerated renal fibrosis when subjected to both diabetic and nondiabetic conditions. This worsened fibrosis is associated with aberrant chemokine and cytokine reprogramming, augmented Wnt signaling and suppressed FAO. We conclude that endothelial GR is a key molecule involved in the regulation of fibrotic processes in the kidney.

## Results

**Endothelial GR deficiency results in a fibrogenic phenotype in the kidneys of diabetic mice.** The streptozotocin (STZ)-induced diabetic CD-1 mouse is the established mouse model to study diabetic kidney disease[29–31], as the kidney fibrosis phenotype is dependent upon mouse strain specificity[30]. Though STZ-induced diabetic CD-1 mice and diabetic C57BL/6 mice demonstrate similar blood glucose levels, the kidneys of diabetic CD-1 mice have been shown to have higher rates of EndMT and more severe fibrosis when compared to the kidneys of diabetic C57BL/6 mice[29,32]. Therefore, diabetic CD-1 mice are considered a profibrotic strain, while diabetic C57B/L6 mice are considered to be a less-fibrotic strain[32,33].

Diabetic CD-1 mouse kidneys displayed significant suppression of GR compared to those from diabetic C57BL/6 mice as assessed by immunofluorescent staining (Fig. 1a). Moreover, ECs isolated from the kidneys of diabetic CD-1 mice showed dramatic suppression in both the GR protein and mRNA levels when compared to the diabetic C57BL/6 mice and the nondiabetic controls of both genotypes (Fig. 1b). The effect of corticosterone deficiency on the progression of kidney fibrosis was evaluated by pursuing bilateral adrenalectomy in both strains of mice. The efficacy of the surgical procedure was verified by substantially suppressed corticosterone levels in all mice studied (Supplementary Fig. 1a); adrenalectomy did not affect glycemia (Supplementary Fig. 1b). Adrenalectomy did not cause any significant difference in either the extent of fibrosis or the collagen expression level (Supplementary Fig. 1c), suggesting that the global suppression of corticosterone did not influence the observed fibrosis phenotype.

**Loss of EC GR worsens kidney fibrosis.** To verify efficient GR excision from ECs in the kidneys of endothelial GR knockout (GR^ECKO) mice, we performed Western blot and qPCR for GR in isolated kidney ECs. Flow cytometry analysis demonstrated that kidney EC isolation was accomplished with ~95% purity (Supplementary Fig. 2). As shown in Supplementary Fig. 3a,b, mRNA and protein levels were significantly diminished in ECs from GR^ECKO mice, as expected. Diabetes was produced by injecting five consecutive low doses of STZ (50 mg/kg/day intraperitoneal (IP)) in 8-week-old GR^ECKO (GR;fl/fl Tie1 Cre+) and Cre− littermate controls (GR^fl/fl) and GR;fl/flTie1 Cre+/Apoe^−/− (DKO) mice and Cre− littermates (GR;fl/fl Apoe^−/−) (Fig. 2a). Animals were monitored for 4 months post STZ treatment before sacrifice. At the time of sacrifice, nondiabetic and diabetic GR^ECKO and DKO mice and their littermate controls had no significant change in body weight, blood glucose, heart weight, liver weight, triglycerides, or cholesterol; however, diabetic GR^EC KO and diabetic

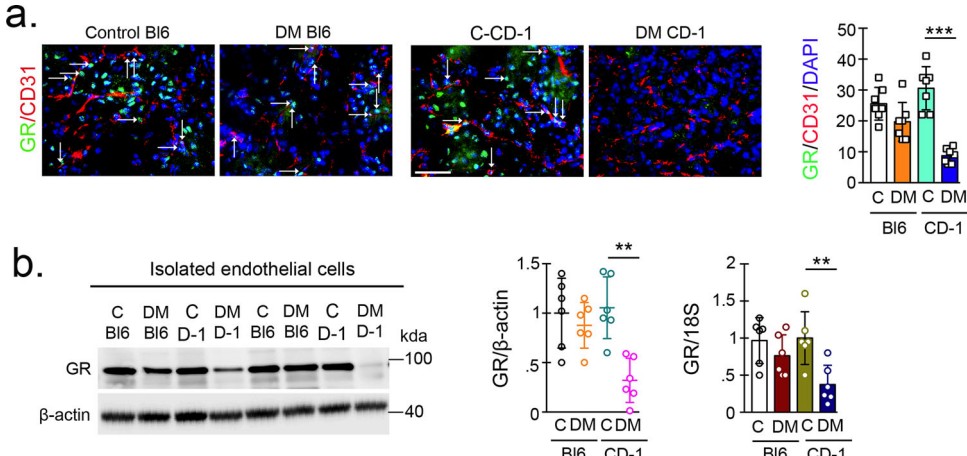

**Fig. 1 Loss of endothelial GR results in a fibrogenic phenotype in the kidneys of diabetic mice. a** Immunofluorescence analysis was performed in the kidneys of control and diabetic CD-1 and C57BL/6 mice using FITC-labeled GR, rhodamine-labeled CD31 and DAPI (blue). Representative merged images are shown; original magnification ×400. Scale bar 50 μm. N = 7 biologically independent mice/group combined from three separate experiments. Quantification of GR/CD31 double-labeled cells is shown. Data are mean ± SEM. **b** Western blot and qPCR analysis of GR protein and mRNA levels in isolated endothelial cells from the kidneys of control and diabetic CD-1 and C57BL/6 mice. Endothelial cells were isolated and cultured from six biologically independent mice/group. Three independent experiments were performed. Densitometry analysis of six samples/group is shown and is normalized to β-actin. mRNA expression from six independent cell culture samples was analyzed and normalized to 18S. Two independent experiments were performed in triplicate. Data are mean ± SEM. For all panels, two-way ANOVA was used for the analysis of statistical significance. Significance *$p < 0.05$, **$p < 0.01$, ***$p < 0.001$. Source data are provided in the source data file.

DKO had relatively higher kidney weight and albumin-to-creatinine ratios when compared to their respective diabetic controls (Fig. 2b–i and Supplementary Fig. 4). The systolic blood pressure of diabetic mice of all genotypes was ~20 mm Hg lower than the corresponding nondiabetic controls, presumably due to the massive polyuria induced by diabetes, but not significantly different otherwise (Supplementary Fig. 4). Diabetic DKO had significantly higher kidney weight and albumin-to-creatinine ratios when compared to diabetic GR$^{EC\ KO}$. Renal fibrosis was assessed by histologic analysis of kidney sections from all genotypes. Diabetic GR$^{ECKO}$ mice exhibited a higher relative area of fibrosis, higher relative collagen deposition and more severe glomerulosclerosis at the 4-month timepoint when compared to diabetic littermate controls (Fig. 2j and Supplementary Fig. 5a). Diabetic DKO exhibited greatly increased relative area of fibrosis and relative collagen deposition when compared to diabetic Apoe$^{-/-}$ controls and diabetic GR$^{ECKO}$ (Fig. 2j). Diabetic GR$^{ECKO}$ mice exhibited higher expression of vimentin, snail1, and HIF1α and lower expression of PPARα at the 4-month timepoint when compared to diabetic littermate controls (Supplementary Figs. 5b and 6). A similar pattern was observed in diabetic DKO when compared to diabetic Apoe$^{-/-}$ controls (Supplementary Figs. 5b and 6). Immunofluorescence data showed higher collagen I and fibronectin deposition in the kidneys of diabetic animals with GR$^{ECKO}$, with the highest deposition observed in DKO mice (Supplementary Fig. 7).

In order to test the role of endothelial GR in nondiabetic fibrosis, UUO in 8-week-old GR$^{ECKO}$ and control littermates was performed (Supplementary Fig. 8a). There was no significant difference in renal fibrosis between contralateral kidneys of controls and GR$^{ECKO}$ mice. However, UUO kidneys from GR$^{ECKO}$ mice showed a greater relative area of fibrosis and greater collagen deposition when compared to UUO kidneys of littermate controls (Supplementary Fig. 8b). Immunofluorescence staining revealed higher collagen I, αSMA, and fibronectin deposition in the UUO kidneys of GR$^{ECKO}$ when compared to UUO kidneys of control littermates (Supplementary Fig. 8c). Administration of the GR agonist dexamethasone abolished the renal fibrogenic phenotype in C57BL/6 mice subjected to UUO by

day 10 (Supplementary Fig. 9a). Dexamethasone treatment slightly suppressed the renal fibrosis in contralateral kidneys of C57BL/6 mice at day 15 but did not cause any remarkable alteration in nondiabetic control CD-1 mice (Supplementary Fig. 9a, b).

**Endothelial GR loss reprograms cytokine and chemokine homeostasis.** Inflammation is a key factor during the fibroblast activation process in the kidneys of diabetic mice[34,35] and disruption of cytokine and chemokine homeostasis contributes to the development of diabetic kidney disease[36–38]. To investigate whether there were derangements in homeostasis in our model, cytokine analysis in the plasma of diabetic mice with severe fibrosis (diabetic CD-1) and the plasma of diabetic mice with less severe fibrosis (diabetic C57BL/6) was performed. Diabetic CD-1 mice demonstrated higher levels of plasma IL-1β, IL-6, IL-10, IL-17, G-CSF, IFN-γ, TNF-α, MCP-1, CCL3, and CCL4 levels, however, the CCL5 level was suppressed when compared to that of diabetic C57BL/6 mice (Supplementary Fig. 10). The same cytokines were also analyzed in the plasma from diabetic and nondiabetic GR$^{ECKO}$ mice and littermate controls and diabetic and nondiabetic DKO and Apoe$^{-/-}$ controls. A similar pattern was observed in both genotypes in which IL-1β, IL-6, IL-10, eotaxin, G-CSF, and CCL4 were significantly higher, while CCL5 was significantly lower, in the plasma of diabetic GR$^{ECKO}$ and diabetic DKO mice when compared to the plasma of their respective diabetic control littermates (Fig. 3a and Supplementary Fig. 11a). Similarly, mRNA gene expression analysis demonstrated that the level of IL-1β, IL-6, IL-10, IL-17, eotaxin, and CCL4 was significantly upregulated, while CCL5 was significantly downregulated, in the kidneys of diabetic GR$^{ECKO}$ and diabetic DKO mice when compared to the diabetic kidneys of their respective control littermates (Supplementary Fig. 11b), indicating more EC inflammation in mice lacking endothelial GR.

Stimulation of isolated kidney ECs from CD-1 mice with recombinant IL-1β, IL-6, TNFα, and TGFβ caused a significant increase in expression of the mesenchymal markers vimentin and

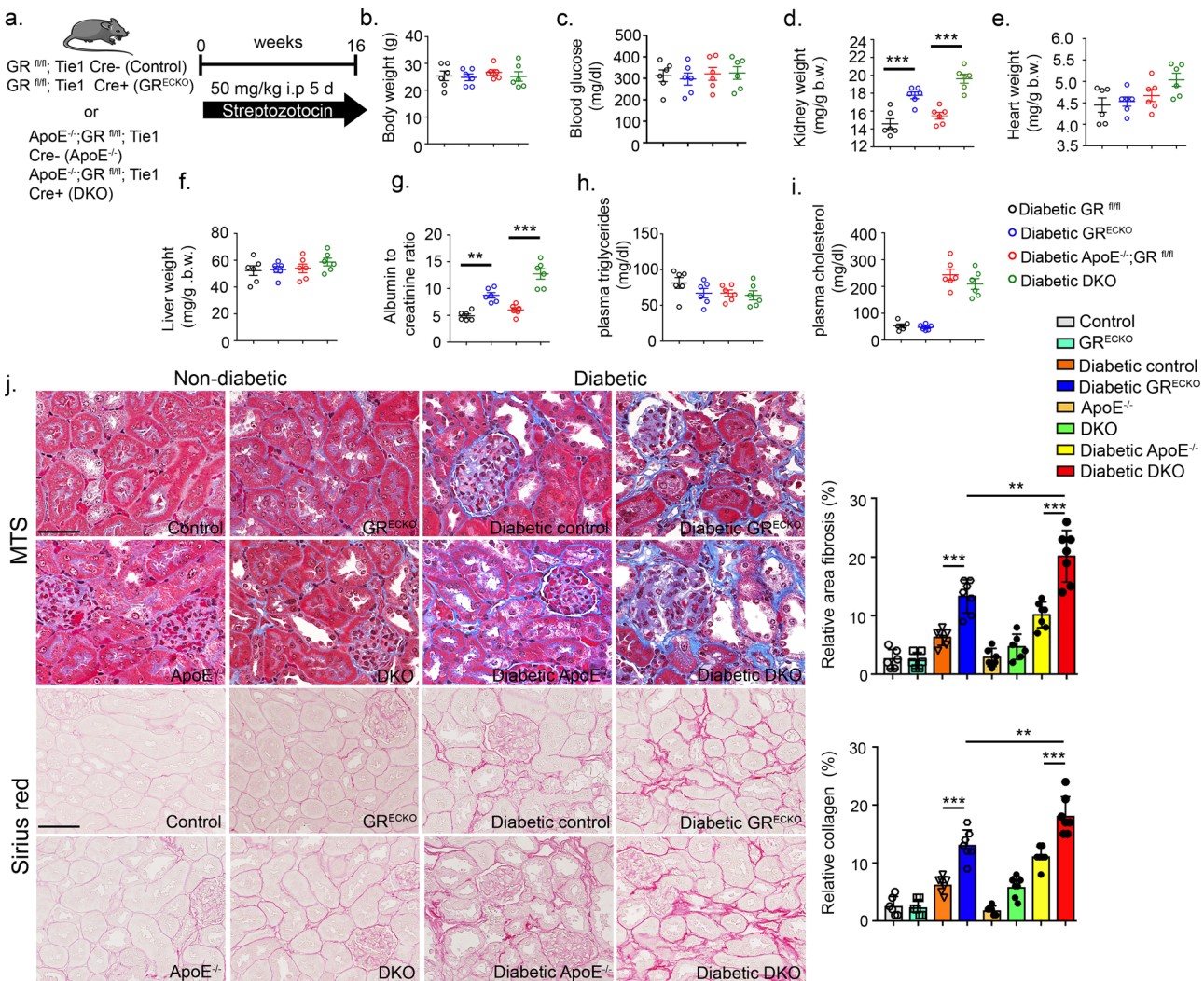

**Fig. 2 Loss of endothelial GR worsens renal fibrosis in diabetic mice. a** Schematic diagram, showing induction of diabetes in *GR fl/fl; Tie1 Cre−* (Control). *GR fl/fl; Tie1 Cre+* (GR^ECKO), *Apoe;−/−GR fl/fl; Tie1 Cre− (Apoe−/−)* and *Apoe;−/−GR fl/fl; Tie1 Cre+* (DKO) mice. Five doses of STZ (50 mg/kg/day IP) were injected to induce diabetes and renal fibrosis. **b–i** Physiological parameters including body weight, blood glucose, kidney weight/body weight, heart weight/body weight, liver weight/body weight, albumin-to-creatinine ratio (ACR), plasma triglycerides, and plasma cholesterol were measured. $N = 6$ biologically independent mice/group from two separate experiments. **j** Masson trichome and Sirius red staining in kidneys of nondiabetic and diabetic control, GR^ECKO, *Apoe−/−*, and DKO mice were analyzed. Representative images are shown; original magnification ×300. Relative area of fibrosis (%) and relative collagen (%) were measured using the ImageJ program. $N = 7$ biologically independent mice/group from two separate experiments. Scale bar 50 μm. Data are mean ± SEM. For all panels, two-way ANOVA was used for the analysis of statistical significance. Significance $*p < 0.05$, $**p < 0.01$, $***p < 0.001$. Source data are provided in the source data file. Components of this figure were created using Servier Medical Art templates, which are licensed under a Creative Commons Attribution 3.0 Unported License; https://creativecommons.org/licenses/by/3.0/.

αSMA (Supplementary Fig. 12a, b); neutralization of IL-6 using IL-6 neutralizing antibody could suppress vimentin expression in TGFβ2-treated and untreated cells (Supplementary Fig. 12c). To further investigate the role of IL-6 in diabetes-associated renal fibrosis, we injected (IP) IL-6 neutralizing antibody in nondiabetic and diabetic CD-1 mice. In nondiabetic mice, IL-6 neutralization did not cause any significant difference; however, in diabetic mice, IL-6 neutralization resulted in a significant reduction of fibrosis and collagen deposition (Supplementary Fig. 12d). In addition, isolated cells from diabetic GR^ECKO and DKO mice had higher IL-6 levels in culture media when compared to isolated cells from their respective diabetic control littermates (Fig. 3b). When IL-6 neutralization was performed in nondiabetic and diabetic GR^ECKO and DKO mice, it completely reversed the fibrogenic phenotype in diabetic animals when compared to respective IgG control-injected mice (Fig. 3c). There

was no significant effect observed in nondiabetic mice (Supplementary Fig. 13).

**Canonical Wnt signaling is a new drug target for the action of endothelial GR.** Given the recently described regulation of Wnt signaling by endothelial GR[27] as well as the recognized role of Wnt signaling in renal fibrosis[28], we assessed the mRNA expression of Wnt-dependent genes and fibrogenic markers in ECs isolated from the kidneys of diabetic GR^ECKO and diabetic DKO mice and their diabetic littermate controls. The expression of Wnt-dependent genes and fibrogenic markers was upregulated in kidneys of diabetic GR^ECKO and diabetic DKO when compared to their respective controls. However, the kidneys of diabetic DKO mice showed the highest expression of both Wnt-dependent genes, such as *axin2* and *tcf*, and fibrogenic markers,

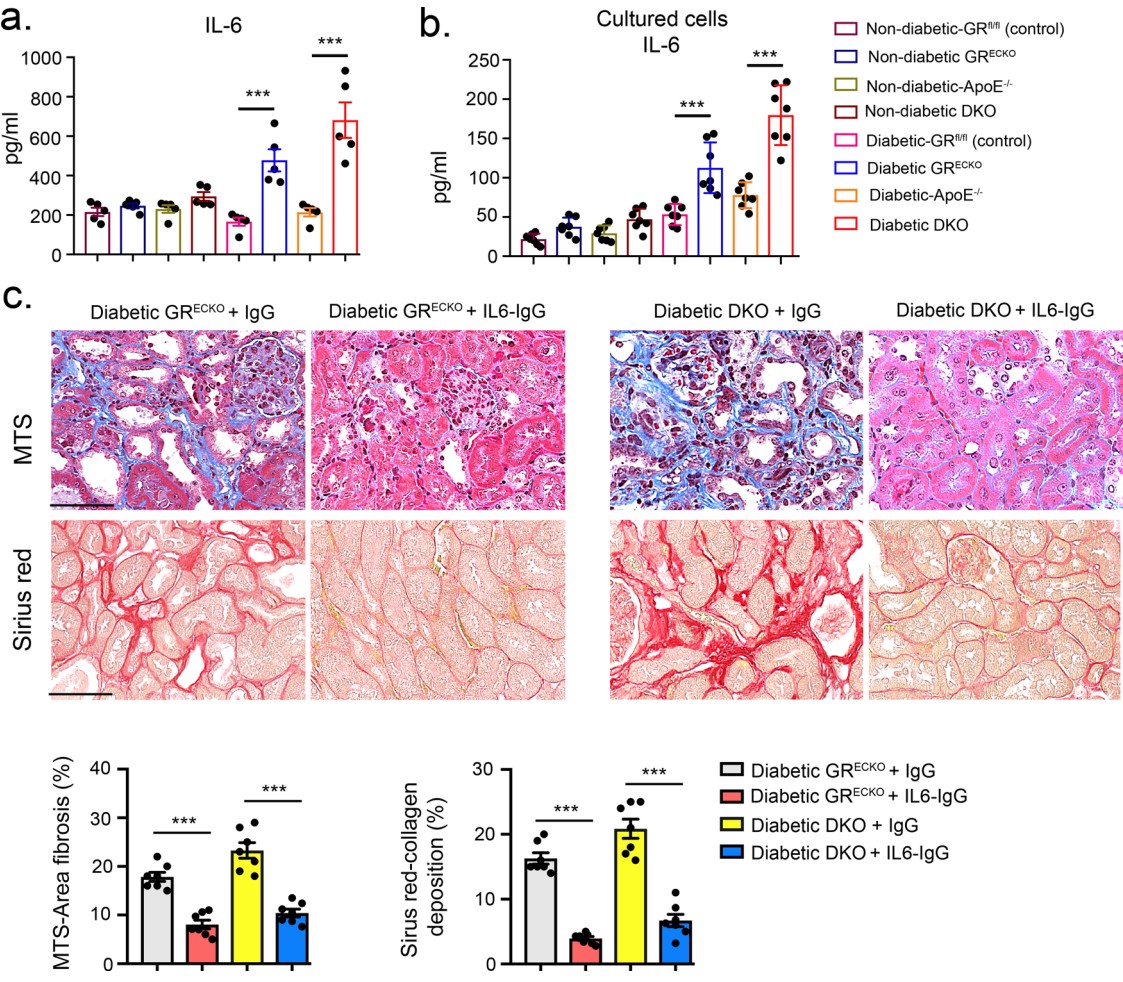

**Fig. 3 IL-6 neutralization rescues renal fibrosis in diabetes. a** Plasma IL-6 levels in nondiabetic and diabetic mice were measured by cytokine array (Luminex). $N = 5$ biologically independent mice/group. **b** Measurement of IL-6 in media of cultured endothelial cells isolated from the indicated groups. Endothelial cells were isolated and cultured from six biologically independent mice per group and were analyzed in triplicate. Data are mean ± SEM. **c** IL-6 neutralization in diabetic GR$^{ECKO}$ and diabetic DKO mice. Representative images are shown; original magnification ×300. Relative area of fibrosis (%) and relative collagen (%) were measured using the ImageJ program. $N = 7$ biologically independent mice/group. Scale bar 50 μm. Data are mean ± SEM. For all panels, one-way ANOVA with Tukey post-test was used for the analysis of statistical significance. *$p < 0.05$, **$p < 0.01$, ***$p < 0.001$. Source data are provided in the source data file.

such as *αSMA* and *fibronectin* (Supplementary Fig. 14a). The expression of *HIF1α* and *Snaill1* was upregulated, while *PPARα* expression was downregulated in ECs isolated from the kidneys of diabetic GR$^{ECKO}$ and diabetic DKO mice when compared to diabetic controls (Supplementary Fig. 14a). In ECs isolated from cultured glomeruli, the expression of Wnt-dependent genes *axin2* and *tcf* and *fibronectin* was upregulated in those from the kidneys of diabetic GR$^{ECKO}$ mice compared to littermate control mice and nondiabetic mice (Supplementary Fig. 14b). Protein expression of β-catenin was also higher in the interstitial and glomerular compartments of diabetic GR$^{ECKO}$ and diabetic DKO compared to their respective controls (Supplementary Fig. 14c). Western blotting demonstrated higher protein expression of vimentin, HIF1α, snail1, and active β-catenin and decreased levels of PPARα and CPT1a in isolated ECs from diabetic GR$^{ECKO}$ when compared to diabetic control littermates (Fig. 4a). Isolated ECs from diabetic DKO mice exhibited higher fibronectin, αSMA, and active β-catenin levels when compared to littermate *Apoe*$^{−/−}$ and diabetic GR$^{ECKO}$ mice (Fig. 4b).

Using immunofluorescent co-staining, the same pattern was also observed, with diabetic GR$^{ECKO}$ and diabetic DKO mice demonstrating higher levels of αSMA/CD31 and TGFβR1/CD31

co-staining in the kidneys when compared to their respective controls, suggestive of EndMT (Fig. 4c). There was no difference in the EC density of ECs from the kidneys of diabetic GR$^{ECKO}$ and diabetic DKO when compared to control ECs; however, EC permeability, as demonstrated by FITC-dextran staining, was higher in kidneys and isolated cultured ECs from diabetic GR$^{ECKO}$ and diabetic DKO when compared to their respective control littermates (Supplementary Fig. 15a–c).

**Inhibition of canonical Wnt signaling improves renal fibrosis.**
Diabetic CD-1 mouse kidneys displayed significantly higher expression of β-catenin, a marker of canonical Wnt signaling, compared to those from diabetic C57BL/6 mice as assessed by immunohistochemical staining (Supplementary Fig. 16a). The kidneys of diabetic CD-1 mice also showed dramatically higher mRNA expression of Wnt-dependent genes and fibronectin when compared to those of the diabetic C57BL/6 mice and the non-diabetic controls of both genotypes (Supplementary Fig. 16b). To determine whether inhibition of the Wnt signaling pathway could ameliorate the observed fibrosis, we utilized LGK974, a small molecule inhibitor of all secreted Wnts[39]. Supplementary Fig. 16c,

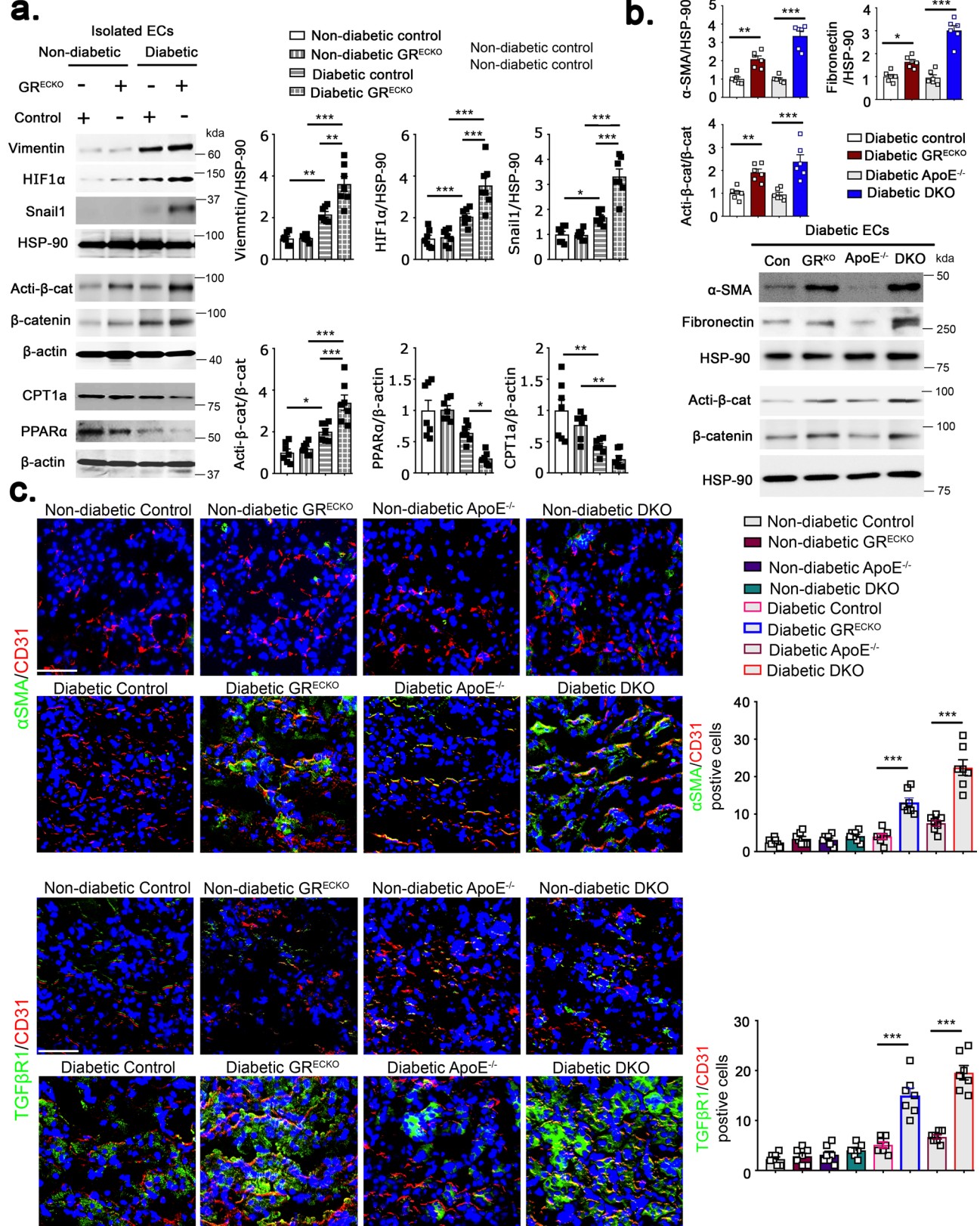

d depicts the schematic diagram showing the experimental protocol for LGK974 treatment in diabetic CD-1 and UUO mice. LGK974 greatly diminished the ECM deposition, relative area of fibrosis, collagen accumulation, and glomerulosclerosis in both models used (Supplementary Fig. 16e, f). Wnt inhibition also substantially restored endothelial GR and suppressed the level of β-catenin in diabetic and UUO mice (Supplementary Fig. 16g–j). In diabetic CD-1 mice, LGK974 also significantly suppressed elevated levels of IL-1β, IL-6, IL-10, G-CSF, TNFα, MCP-1, and CCL4 while increasing the level of CCL5 (Supplementary Fig. 17).

**Fig. 4 Upregulation of Wnt signaling and fibrogenic markers with loss of endothelial GR. a** Western blot analysis of vimentin, HIF1α, Snail1, active β-catenin, CPT1a, and PPARα in isolated endothelial cells from the kidneys of nondiabetic and diabetic control and GR$^{ECKO}$ mice. Endothelial cells were isolated and cultured from six biologically independent mice/group. Densitometry calculations from six samples/group combined from three independent experiments are shown and are normalized to β-actin or HSP90 as indicated. Representative blots are shown. Data are mean ± SEM. **b** Western blot analysis of αSMA, fibronectin, active β-catenin and β-catenin in isolated endothelial cells from the kidneys of diabetic control, GR$^{ECKO}$, Apoe$^{-/-}$, and DKO mice. Endothelial cells were isolated and cultured from six biologically independent mice/group. Densitometry calculations from six samples/group combined from three independent experiments are shown and are normalized to β-actin or HSP90 as indicated. Data are mean ± SEM. Representative blots are shown. **c** Immunofluorescence analysis of αSMA/CD31 and TGFβR1/CD31 was performed in the kidneys of nondiabetic and diabetic control, GR$^{ECKO}$, Apoe$^{-/-}$, and DKO mice. FITC-labeled αSMA, FITC-labeled TGFβR1, rhodamine-labeled CD31 and DAPI (nuclei, blue) were used. Representative merged images are shown; original magnification ×300. Scale bar 50 μm. N = 7 biologically independent mice/group. Data are mean ± SEM. For all panels, one-way ANOVA with Tukey post-test was used for the analysis of statistical significance. Significance *p < 0.05, **p < 0.01, ***p < 0.001. Source data are provided in the source data file.

**Wnt inhibitor partially suppresses the fibrogenic phenotype in the kidneys of diabetic GR$^{ECKO}$.** To determine whether Wnt inhibition could mitigate the renal fibrosis observed in diabetic GR$^{ECKO}$ mice, a cohort of animals was treated with LGK974. At the age of 8 weeks, control and GR$^{ECKO}$ mice were injected with STZ 50 mg/kg for 5 consecutive days. Sixteen weeks after injection, LGK974 was administered by oral gavage for 8 additional weeks (Fig. 5a). At the time of sacrifice, there were no significant differences in body weight or blood glucose among the groups (Fig. 5b and Supplementary Fig. 18). However, a significant reduction in kidney weight was observed in the Wnt inhibitor-treated diabetic GR$^{ECKO}$ and diabetic control mice (Fig. 5b). Wnt inhibition clearly improved the relative area of fibrosis, relative collagen deposition, and tubular damage in diabetic control mice; this effect was less pronounced, though still significant in diabetic GR$^{ECKO}$ mice (Fig. 5c). Wnt inhibition did not result in any remarkable change in the renal fibrogenic phenotype or β-catenin level in nondiabetic control mice or nondiabetic GR$^{ECKO}$ when compared to their untreated nondiabetic controls (Supplementary Fig. 19). Wnt inhibition suppressed the expression of Snail1 and HIF1α and significantly increased the level of PPARα in the kidneys of diabetic GR$^{ECKO}$ (Supplementary Fig. S20). A similar effect was observed in isolated ECs (Supplementary Fig. 21). Wnt inhibition also significantly suppressed EndMT (CD31/αSMA double-positive cells) in diabetic control mice; this effect was less pronounced in diabetic GR$^{ECKO}$ mice (Supplementary Fig. 22). However, Wnt inhibition significantly reduced the level of EMT (E-cadherin/αSMA double-positive cells) in control and diabetic GR$^{ECKO}$ mice (Supplementary Fig. 22).

**Metabolic reprogramming by loss of endothelial GR accelerates renal fibrosis.** It is increasingly recognized that defects in central metabolism contribute to kidney fibrosis[32,40]. Defective FA metabolism in ECs leads to EndMT events[41]. To investigate whether FA metabolism was deranged in our model, radiolabeled [14C]palmitate uptake experiments in isolated ECs from mouse kidneys were performed. We observed that FA uptake was higher in isolated ECs from diabetic kidneys of the more fibrotic strain (diabetic CD-1) when compared to kidney ECs from the less-fibrotic strain (diabetic C57BL/6). Administration of the Wnt inhibitor suppressed FA uptake in ECs from CD-1 mice (Supplementary Fig. 23a). Kidney ECs from both diabetic GR$^{ECKO}$ and DKO mice displayed higher FA uptake when compared to those of the diabetic control littermates (Fig. 6a). FAO was also assessed by measuring the [14CO$_2$]release from radiolabeled [14C]palmitate in cultured ECs isolated from kidneys. FAO was diminished in the isolated kidney ECs of diabetic CD-1 mice and Wnt inhibition was able to restore the level of FAO (Supplementary Fig. 23b). Cultured kidney ECs from diabetic GR$^{ECKO}$ and diabetic DKO mice showed a diminished level of FAO when compared to their diabetic control littermates (Fig. 6b). In addition, gene expression of FA transporter proteins 1 and 4 (FATP1

and FATP4) were unaltered in both nondiabetic and diabetic ECs isolated from kidneys of control and GR$^{ECKO}$ mice; however, the gene expression level of CD36, another FA transporter, was lower in diabetic mice of both genotypes when compared to nondiabetic controls (Supplementary Fig. 24).

In the next set of experiments, diabetic CD-1 mice were treated with the FA synthase inhibitor C75, the FAO inhibitor etomoxir, the PPARα agonist fenofibrate, or the cholesterol-lowering drug simvastatin for 4 weeks. Fenofibrate and C75 ameliorated the fibrogenic phenotype, whereas etomoxir exacerbated the fibrosis. Simvastatin treatment did not cause significant suppression in the level of fibrosis (Supplementary Fig. 25a). Fenofibrate and C75 restored the level of GR protein in CD31 positive cells, whereas etomoxir suppressed it and simvastatin did not show any substantial effect (Supplementary Fig. 25b). Fenofibrate and C75 downregulated fibronectin and αSMA mRNA levels, while etomoxir upregulated this mRNA and simvastatin did not cause any significant change in the gene expression level of fibronectin or αSMA in the diabetic kidneys (Supplementary Fig. 25c). These FA modulators did not cause any significant differences in the blood glucose levels (Supplementary Fig. 25d). Etomoxir treatment caused significant suppression of FAO, as measured by [14CO$_2$]release, and CPT1a level, and induced the expression of β-catenin, whereas C75 and fenofibrate increased the level of FAO, increased the expression of CPT1a, and suppressed the level of β-catenin in diabetic CD-1 mice (Supplementary Fig. 25e, f).

Etomoxir and C75 were also tested in nondiabetic and diabetic control littermates and GR$^{ECKO}$ mice (Fig. 6c, d and Supplementary Figs. 26 and 27). There were no significant differences in body weight, blood glucose, or kidney weight in nondiabetic or diabetic control littermates and GR$^{ECKO}$ mice after treatment with etomoxir or C75 (Fig. 6c and Supplementary Fig. 26). Data from kidney ECs revealed that etomoxir caused significant suppression of FAO, and C75 restored FAO in nondiabetic control littermates (Supplementary Fig. 26). However, in kidney ECs from diabetic GR$^{ECKO}$ mice, etomoxir caused significant suppression of FAO which C75 was unable to rescue (Fig. 6c). Etomoxir treatment accelerated the renal fibrogenic phenotype, suppressed the CPT1a level and increased the expression level of β-catenin in the kidneys of diabetic control and diabetic GR$^{ECKO}$ mice. C75 treatment clearly abolished the renal fibrogenic phenotype, restored CPT1a, and completely diminished the level of β-catenin in the kidneys of diabetic control mice. These effects were also observed in the GR$^{ECKO}$ mice, though to a lesser extent (Fig. 6d). Etomoxir slightly increased the level of renal fibrosis in nondiabetic control and GR$^{ECKO}$ mice, which could be mitigated by C75 (Supplementary Fig. 27). Etomoxir decreased expression of CPT1a in diabetic control mice while also increasing expression of β-catenin; C75 was able to substantially reverse both of these effects. A similar pattern was observed in diabetic GR$^{ECKO}$ mice, though again, to a lesser extent (Fig. 6e).

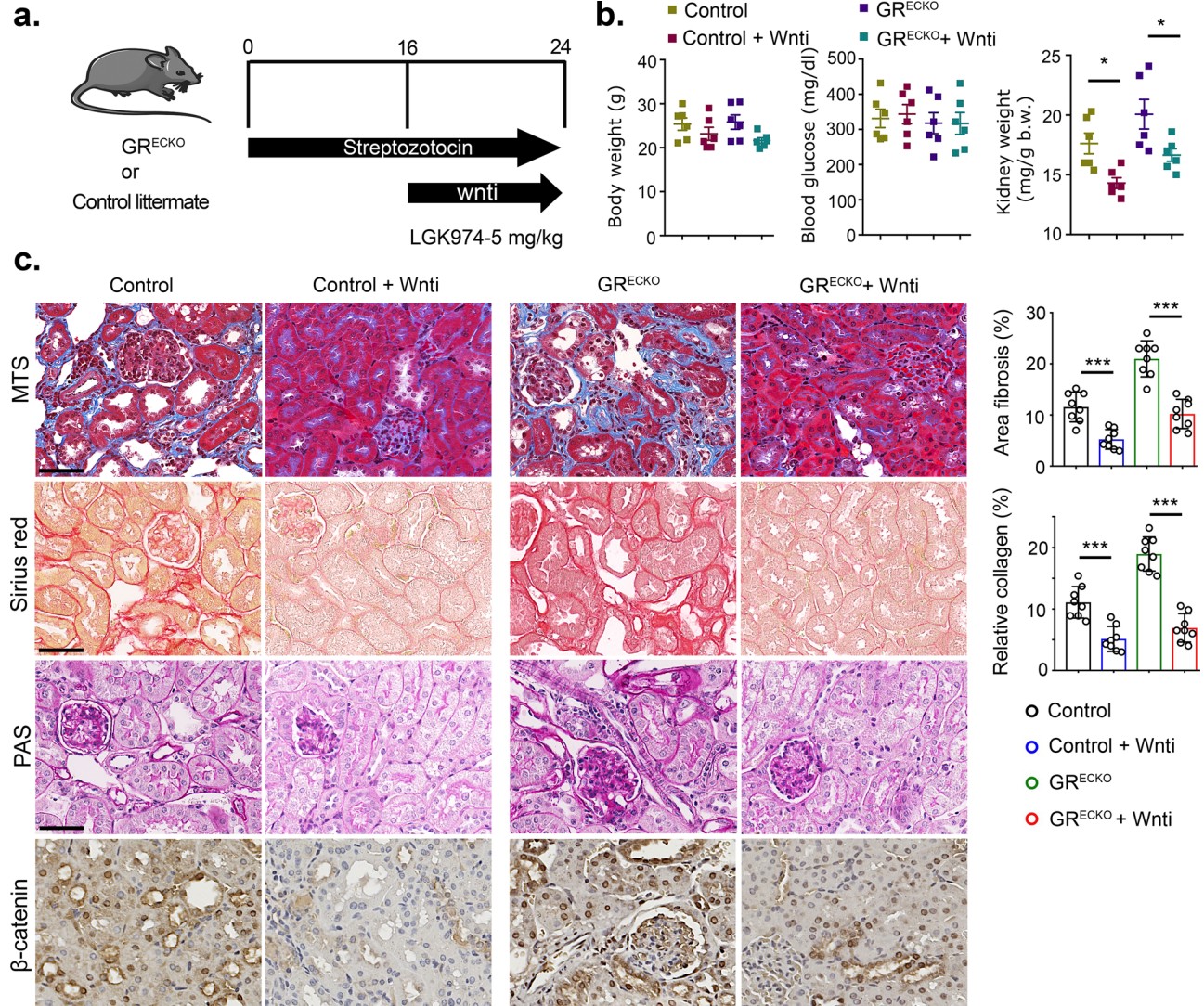

**Fig. 5 Wnt inhibitor partially abrogates renal fibrosis in diabetic GR^ECKO and DKO mice. a** Schematic diagram showing the treatment protocol of Wnt inhibition in diabetic control and GR^ECKO mice. **b** Physiological parameters including body weight, blood glucose, and kidney weight/body weight were analyzed. $N = 6$ biologically independent mice/group combined from two experiments. **c** Masson trichrome, Sirius red, and PAS staining as well as immunohistochemical analysis of β-catenin levels in the kidneys of wnti-treated diabetic control and GR^ECKO mice. Representative images are shown; original magnification ×300. Relative area of fibrosis (%) and relative collagen deposition (%) were measured using the ImageJ program. $N = 8$ biologically independent mice/group combined from three experiments were analyzed. Scale bar 50 µm. Data are mean ± SEM. For all panels, one-way ANOVA with Tukey post-test was used for the analysis of statistical significance. Significance *$p < 0.05$, **$p < 0.01$, ***$p < 0.001$. Source data are provided as source data file. Components of this figure were created using Servier Medical Art templates, which are licensed under a Creative Commons Attribution 3.0 Unported License; https://smart.servier.com.

**GR loss-linked EndMT disrupts central metabolism and induces mesenchymal transformation in tubular epithelial cells**. These in vivo data suggest that GR^ECKO mice exhibit enhanced EndMT and EMT in their diabetic kidneys. Over-expression of GR suppressed the TGFβ2-stimulated increase in αSMA, collagen I, and β-catenin expression in HUVECs (Supplementary Fig. 28). To test whether endothelial GR deficiency affects mesenchymal programs and causes defects in central metabolism in neighboring epithelial cells, we analyzed the effects of conditioned media (CM) from GR knockdown HUVECs on the mesenchymal phenotype of HK-2 cells (Fig. 7a). CM from GR siRNA-transfected HUVECs decreased E-cadherin protein levels and increased αSMA, TGFβR1, and β-catenin protein levels in HK-2 cells when compared to media from scrambled siRNA-transfected HUVECs (Fig. 7b, c). CM treatment from GR siRNA-transfected HUVECs caused a significant reduction in the level of

FAO, oxygen consumption rate (OCR), and cellular ATP level in HK-2 cells (Fig. 7d–f). TECs incubated with CM from GR siRNA-treated HUVECs exhibited significantly downregulated mRNA expression of the FAO-responsive genes *Cpt1a*, *Cpt2*, *Pparα*, and *Pgc1α* compared to TECs incubated with CM from control siRNA-treated HUVECs (Fig. 7g).

To confirm these in vitro results, primary ECs from diabetic control and diabetic GR^ECKO mice were isolated to analyze the contribution of GR-deficient ECs on mesenchymal activation in TECs. CM from isolated cultured ECs from the kidneys of diabetic GR^ECKO and diabetic control littermates was transferred to cultured TECs from diabetic control mice (Fig. 7h). This CM treatment from GR-deficient cells caused significant suppression of E-cadherin, CPT1a, and PPARα and induced αSMA, TGFβR1, and active β-catenin protein levels in TECs (Fig. 7i). Furthermore, CM from ECs from diabetic GR^ECKO mice significantly

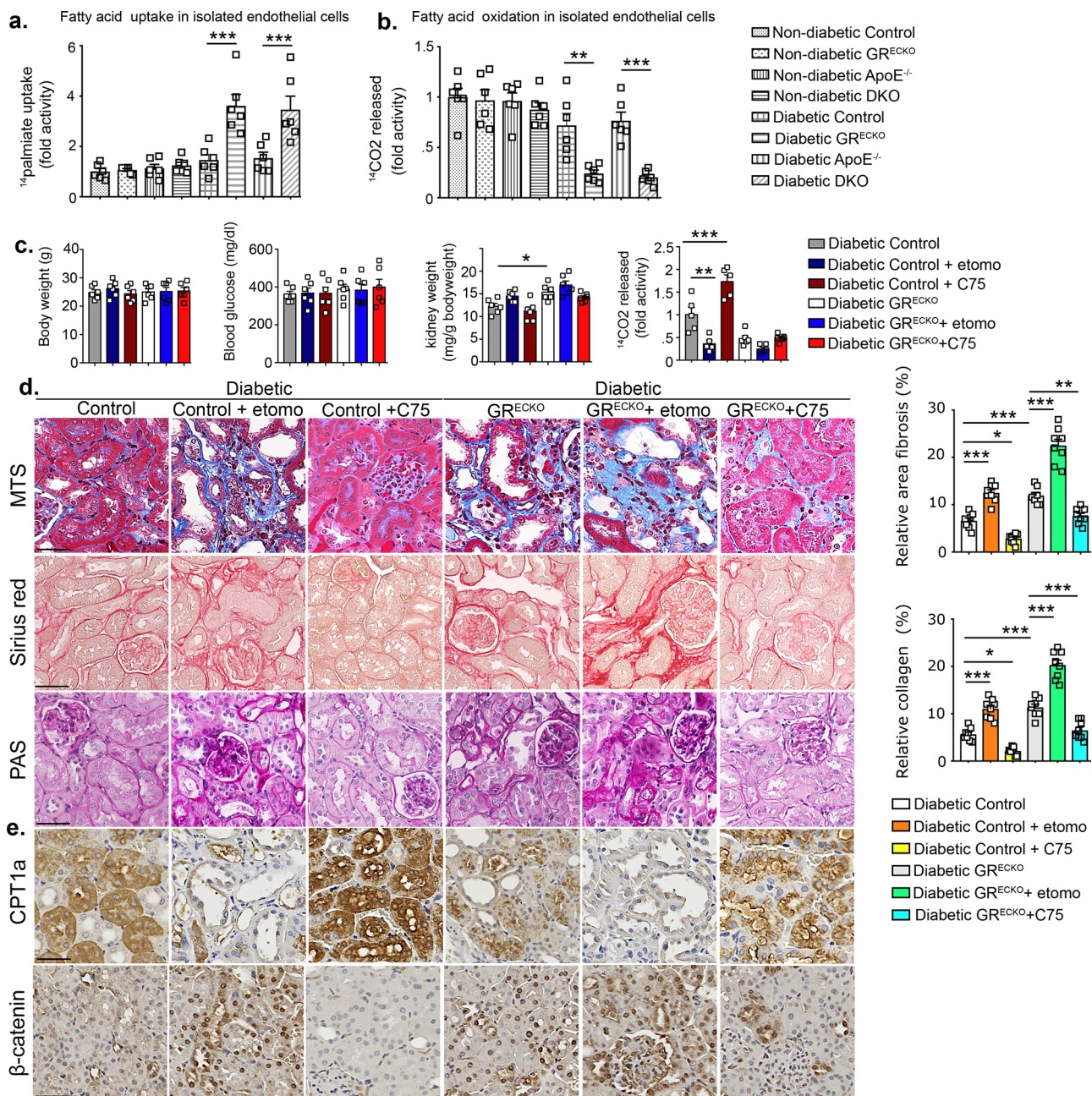

**Fig. 6 Metabolic reprogramming by loss of endothelial GR loss worsens diabetic kidney disease. a** Radiolabeled $^{14}$C-palmitate uptake analysis in isolated endothelial cells from kidneys of nondiabetic and diabetic control, GR$^{ECKO}$, *Apoe$^{-/-}$*, and DKO mice. CPM of each sample were counted. Data were normalized to μg protein. Fold activity is presented. Endothelial cells isolated and cultured from six biologically independent mice/group were performed in triplicate and analyzed for $^{14}$C-palmitate uptake analysis. Data are mean ± SEM. **b** Radiolabeled [$^{14}$C]palmitate oxidation and [$^{14}$CO$_2$]release were measured in isolated endothelial cells from kidneys of nondiabetic and diabetic control, GR$^{ECKO}$, *Apoe$^{-/-}$*, and DKO mice. CPM of each sample was counted. Data were normalized to μg protein. Fold activity is presented. Endothelial cells isolated and cultured from six biologically independent mice/group were performed in triplicate and analyzed for $^{14}$C-palmitate uptake analysis. Data are mean ± SEM. **c** Body weight, blood glucose, and kidney weight/body weight were measured at the end of these experiments. $N = 6$ biologically independent mice from two experiments. Ex vivo radiolabeled [$^{14}$C] palmitate oxidation and [$^{14}$CO$_2$]release were measured in kidneys. CPM of each sample was counted. Isolated endothelial cells from five biologically independent mice/group were analyzed in triplicate. Data are mean ± SEM. **d** Masson trichrome, Sirius red, and PAS staining were analyzed in the kidneys of control-, etomoxir-, and C75-treated diabetic mice. Representative images are shown; original magnification ×300. Relative area of fibrosis (%) and relative collagen deposition (%) were measured using the ImageJ program. $N = 8$ biologically independent mice per/group from three independent experiments were analyzed. Scale bar 50 μm. Data are mean ± SEM. **e** Immunohistochemical analysis of CPT1a and β-catenin. Representative images are shown; original magnification ×300. Scale bar 50 μm. $N = 6$ biologically independent mice per group from three independent experiments. For all panels one-way ANOVA with Tukey post-test was used for the analysis of statistical significance. Significance *$p < 0.05$, **$p < 0.01$, ***$p < 0.001$. Source data are provided in the source data file.

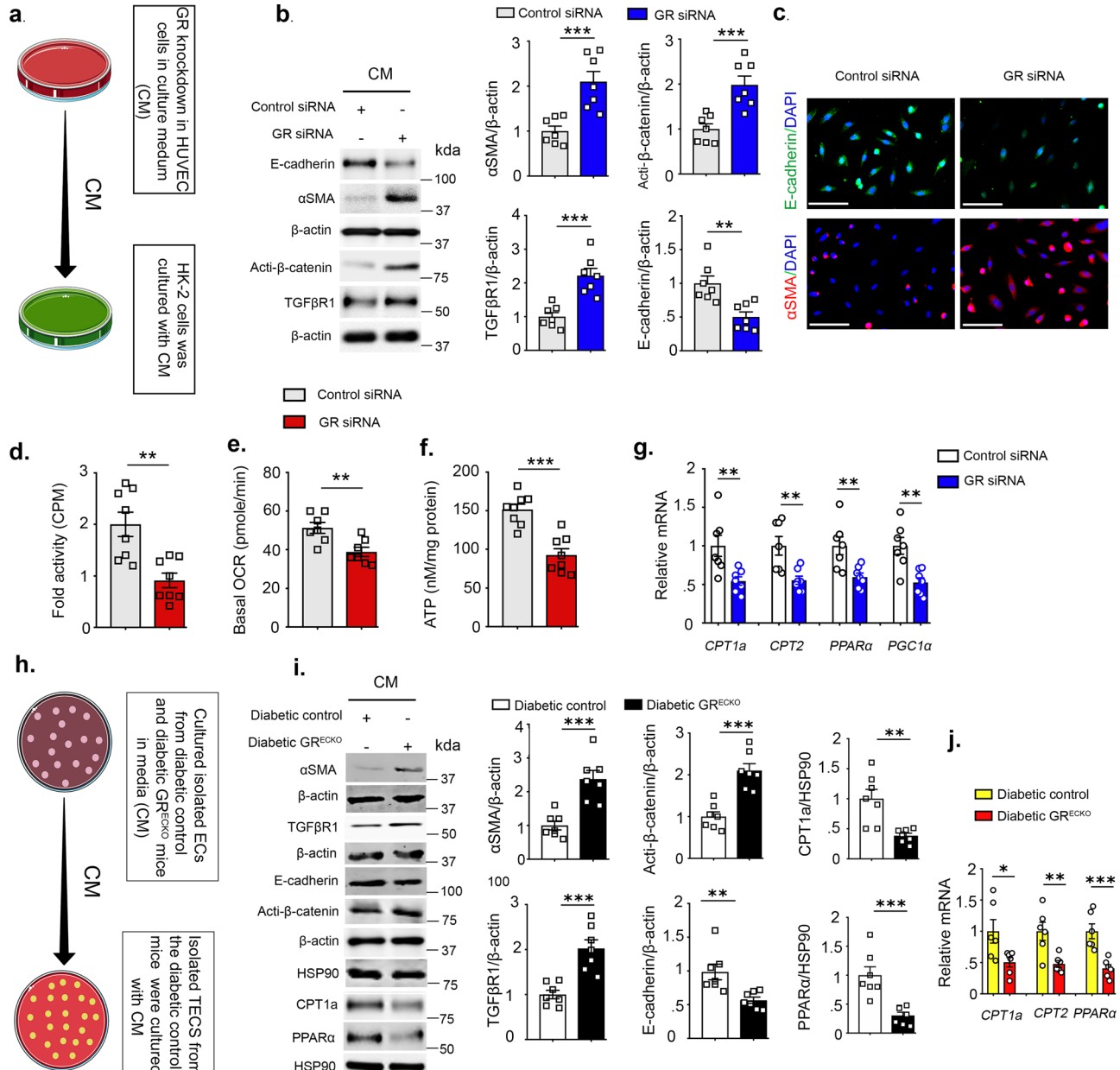

**Fig. 7 GR loss in endothelial cells reprograms central metabolism in renal tubular cells. a** Conditioned media experimental design. HUVECs were transfected with scrambled or GR siRNA; after 6 h, the medium was changed and cells were incubated for 96 h. The subsequently harvested media was transferred to HK-2 cells. **b** Representative Western blotting analysis of E-cadherin, αSMA, TGFβR1, and active β-catenin expression. $N = 7$ independent experiments. Representative blots from seven biologically independent samples/group were combined from four independent experiments. Densitometric analysis relative to β-actin is shown. **c** Immunofluorescence analysis of E-cadherin and α-SMA expression in conditioned medium-treated TECs. For each slide, images of six different fields of view at Å ~ 400 magnification were evaluated. $N = 7$ independent samples/group. Scale bar 30 μm. **d** [$^{14}$C]palmitate oxidation measured by [$^{14}CO_2$]release. CPM were counted and normalized to the protein in the well. $N = 8$ independent samples/group. **e** Oxygen consumption rate (OCR) in conditioned medium-treated TECs; each data-point represents the mean of eight independent samples. Three independent experiments were performed. OCR was measured in a Seahorse XF96 analyzer. **f** Cellular ATP measurement. Eight independent samples were analyzed. **g** Relative mRNA levels determined by qRT-PCR of FAO regulators in conditioned media-treated TECs. $N = 7$ independent samples per group performed in triplicate were analyzed. **h** Experimental design for conditioned media experiment. ECs from the kidneys of diabetic GR$^{ECKO}$ and diabetic control mice were cultured for 96 h. The subsequently harvested media was transferred to TECs from diabetic control mice. $N = 7$/group. **i** Representative Western blotting analysis of αSMA, TGFβR1 active β-catenin, E-cadherin, CPT1a, and PPARα expression. Representative blots from seven biologically independent samples/group combined from three independent experiments are shown. Densitometry calculations are normalized to β-actin or HSP90 as indicated. **j** Relative mRNA levels determined by qRT-PCR of regulators of FAO in conditioned media-treated isolated TECs. $N = 6$ independent replicates per group. Data are mean ± SEM. Student's $t$ test (unpaired two tailed) was used for (**b**–**f**); (**i**); for (**g**, **j**), one-way ANOVA with Tukey post hoc test was used for the analysis of statistical significance. Significance *$p < 0.05$, **$p < 0.01$, ***$p < 0.001$. Source data are provided in the source data file. Components of this figure were created using Servier Medical Art templates, which are licensed under a Creative Commons Attribution 3.0 Unported License; https://smart.servier.com.

downregulated mRNA expression of the FAO-responsive genes *Cpt1a*, *Cpt2*, and *Pparα* (Fig. 7j).

## Discussion

This study demonstrates the crucial role of endothelial GR in the regulation of fibrogenic processes in a mouse model of diabetic kidney disease. Our results demonstrate that endothelial GR regulates the mesenchymal trans differentiation process by influencing FA metabolism and control over canonical Wnt signaling in the kidneys of diabetic mice. It is clear from our data that GR loss is one of the catalysts of renal fibrosis in diabetes that leads to disruption of cytokine and chemokine homeostasis by up regulating canonical Wnt signaling. Ultimately, these processes alter the metabolic switch in favor of defective FA metabolism and associated mesenchymal activation in TECs.

Metabolic reprogramming in ECs is a critical event in the development of myofibroblast formation, proliferation, and fibrosis in diabetic kidneys[12,17,40,42,43]. Our data suggest that GR deficiency is a critical step in the metabolic reprogramming of kidney ECs. While bilateral adrenalectomy suppressed corticosterone significantly in all mice studied, this global loss of systemic steroid signaling was not sufficient to alter the course of fibrosis in diabetic mice, suggesting that the tissue-specific effects of targeted loss of GR in the endothelium supersede the systemic effects; this phenomenon will require further study. The altered cytokine levels in the plasma of GR[ECKO] mice include elevated levels of pro-inflammatory cytokines (IL-1β, IL-6, and IL-17) and the anti-inflammatory cytokine IL-10. The role of IL-10 has not been fully investigated in renal fibrosis in diabetic kidney disease so far. However, our data demonstrate that IL-6 is a key inflammatory cytokine, which is elevated in states of endothelial GR suppression. The neutralization of IL-6 in diabetic mice completely reversed the renal fibrotic phenotype, suggesting a critical pro-fibrotic role of IL-6 in diabetes.

Recently, we demonstrated that loss of endothelial GR results in upregulation of canonical Wnt signaling[27]. It is accepted that GR performs its anti-inflammatory actions by targeting the NFkB signaling pathway[44]. However, GR targets also canonical Wnt signaling in ECs, which is independent of its classic target, NFkB[27,44]. Inhibition of Wnt signaling in ECs may prove to be a valuable therapeutic opportunity for combatting diabetic kidney disease. The Wnt pathway is known to be an important contributor to renal fibrosis and activated canonical Wnt signaling contributes to the disruption of cytokine and chemokine homeostasis[28,45–47]. Our data demonstrate that higher levels of GR-deficient-canonical Wnt signaling are associated with the induction of mesenchymal and fibrogenic markers.

To further test the therapeutic potential of Wnt inhibition, we used the small molecule LGK974. Wnt inhibition clearly suppressed canonical Wnt signaling, substantially improved the fibrogenic phenotype in our mouse model of diabetic kidney disease and restored the endothelial GR level. These data suggest that tonic repression of canonical Wnt signaling in ECs is one mechanism by which GR performs its anti-fibrotic action. Notably, this effect was less evident in GR[ECKO] mice, possibly since Wnt inhibition was able to suppress EMT processes in other cell types (TECs) while it was unable to mitigate EndMT processes. Cumulatively, these data suggest that endothelial GR is a key anti-EndMT molecule.

Research investigating lipid metabolism in kidney cells is limited[48,49] but gaining importance. Defects in central metabolism contribute to diabetic kidney disease[32,40] and there are a few reports showing that altered cytokine levels can affect renal lipid metabolism in diabetic kidney disease[50,51]. Clinical observations indicate a potential association between lipid levels and kidney

disease[52], and lipid control appears to be important in the prevention and treatment of diabetic kidney disease[48,49]. Here, we aimed to dissect the contribution of lipid metabolism in ECs to the regulation of diabetic kidney disease. The connection between atherosclerosis and glomerulosclerosis was suggested two decades ago by Diamond, who introduced the foam cell (lipid overloaded macrophage) as the pivotal culprit in both disease processes[53,54]. The Study of Heart and Renal Protection study was a double-blind, placebo-controlled trial that tried to assess the safety and efficacy of reducing LDL cholesterol in more than 9000 patients, with or without diabetes, with chronic kidney disease[55,56]. Of ~6000 patients who were not on dialysis at randomization, allocation to simvastatin therapy did not produce a significant reduction by any measure of renal disease progression[55]. There are several clinical trials on statins and their effects on kidney outcome[57–60]. These clinical data suggest that lipid lowering drugs improve cardiovascular function in diabetic patients, but do not necessarily improve the renal outcome[49,61]. Large-scale clinical trials that are prospective, randomized, and controlled are still lacking[49].

Based upon our previous work, DKO mice show worsened atherosclerosis, compared to $Apoe^{-/-}$ mice, which is not explained by differences in plasma lipid levels[25]. This observation was the catalyst which led to the evaluation of the diabetic phenotype in these animals. It is clear from our data that hypercholesterolemia worsened the severity of renal fibrosis in GR[ECKO] mice, suggesting that hypercholesterolemia affects EC metabolism and contributes to renal fibrosis. However, similar to available clinical data, the cholesterol-lowering drug simvastatin did not ameliorate the severity of renal fibrosis in this mouse model of diabetic kidney disease. Interestingly, fibrates are a class of drugs that treat hypertriglyceridemia with residual elevation of non-HDL cholesterol. However, the role of fibrates in patients with diabetic kidney disease has yet to be determined[49,62].

We assessed the contribution of endothelial GR loss-linked Wnt activation and its association with defects in FA metabolism. Disruption of endothelial FA metabolism contributes to the activation of EndMT in diabetic kidneys[8,31,63]. FAO activation caused remarkable suppression of fibrosis by restoring the endothelial GR level in diabetic mice. In contrast, FAO inhibition caused acceleration of fibrosis by diminishing the level of endothelial GR in diabetic control mice, suggesting that endothelial GR is a critical protein for the action of FAO modulators. Our data clearly suggest that the anti-fibrotic effect of the FAO activator C75 is dependent on endothelial GR, in turn, suggesting that ECs are required for the anti-fibrotic action of this drug.

When CM from GR-deplete ECs from diabetic GR[ECKO] mice were transferred to cultured TECs from diabetic control kidneys, we observed induction of mesenchymal markers, activation of TGFβ and canonical Wnt signaling, and concomitant suppression of epithelial cell markers. These findings suggest that EndMT leads to the activation of EMT processes in diabetes. GR-deplete cells have higher levels of TGFβ-smad3 and canonical Wnt signaling, which are associated with disrupted levels of plasma cytokines and suppressed FAO. The cumulative effects of these metabolic changes result in the activation of mesenchymal transformation in ECs, which appears to exert paracrine effects on neighboring TECs. The functional importance of GR protein in EC homeostasis is depicted graphically (Fig. 8).

GR agonists like dexamethasone activate GR signaling in all cell types; however, in diabetes, dexamethasone intervention is not preferred due to the severe and predictable exacerbation of hyperglycemia. Alternative approaches which can activate GR in a cell-specific manner need to be identified and may be useful for next generation therapy for cardiovascular dysfunction and renal fibrosis in DN. These data highlight the regulatory role of GR in

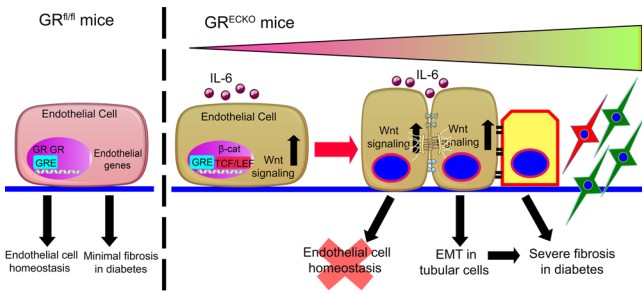

**Fig. 8 Graphical representation of the role of GR in endothelial cell homeostasis.** GR is crucial for endothelial cell homeostasis and its loss causes homeostatic disruption through release of the pro-inflammatory cytokine IL-6 and aberrant activation of Wnt signaling which lead to defective fatty acid oxidation and associated mesenchymal activation. Components of this figure were created using Servier Medical Art templates, which are licensed under a Creative Commons Attribution 3.0 Unported License; https://smart.servier.com.

ECs. However, it is unclear whether endothelial GR directly regulates FAO in mitochondria or indirectly regulates this process by affecting Wnt signaling. What is the potential impact of intracellular GR translocation on the metabolic shift in ECs? It will be interesting to study the role of upstream regulators of GR that might have a significant effect on disease phenotypes in the kidney. Further studies will be required to understand cell-specific metabolic communication in kidney disease pathogenesis.

In conclusion, our findings highlight the regulatory role of GR on EndMT in diabetic kidneys, mediated by control over canonical Wnt signaling and linked defective FA metabolism. This study provides insight into the biology of GR and its critical role in renal fibrosis and diabetes.

## Methods

**Reagents and antibodies**. Rabbit polyclonal anti-GR (SAB4501309), mouse monoclonal anti-αSMA (Cat:A5228), and mouse monoclonal anti-β-actin (AC-74) (A2228) antibodies were from Sigma (St Louis, MO). Anti-TGFβR1 (ab31013), PPARα (ab215270), mouse monoclonal anti-vimentin (RV202) (ab8978), rabbit polyclonal anti-αSMA (ACTA2) (ab5694), anti-HIF1α (ab516008), and goat polyclonal anti-Snail1 (ab53519) antibodies were purchased from Abcam (Cambridge, UK). Mouse anti-β-catenin antibody (610154) was purchased from BD Biosciences. CPT1a (12252), rabbit polyclonal anti-E-cadherin antibody (24E10) (3195), and rabbit non-phospho (active) β-Catenin (8814) antibodies were purchased from Cell Signaling Technology (Danvers, MA, USA). Anti-HSP90 was purchased from BD Biosciences (610419). In vivo mouse IL-6 IgG neutralization antibody and control IgG antibodies were purchased from Bio X Cell. Fluorescence-, Alexa fluor 647-, and rhodamine-conjugated secondary antibodies were obtained from Jackson ImmunoResearch (West Grove, PA). TGFβ2, IL-1β, and recombinant TNFα and TGFβ neutralizing antibodies were purchased from PeproTech (Rocky Hill, NJ). Etomoxir, C75, and Wnt inhibitor (LGK974) were purchased from Cayman Chemical (Ann Arbor MI).

**Animal experimentation**. All experiments were performed according to a protocol approved by the Institutional Animal Care and Use Committee at the Yale University School of Medicine and were in accordance with the National Institute of Health (NIH) Guidelines for the Care of Laboratory Animals. Mice were housed at an ambient temperature of 68–79 °F with a humidity that ranged between 30 and 70%. They were exposed to 12-h light–dark cycles. Mice lacking the endothelial GR (known as GR^ECKO) and those lacking this receptor on the *Apoe* null background (DKO) were generated as previously described[25]; these mice were on the C57BL/6 background. Diabetes was induced in 10-week-old male mice with five consecutive IP doses of STZ 50 mg/kg in 10 mmol/L citrate buffer (pH 4.5). Wnt inhibitor (LGK974) was provided to 16-week-STZ-treated diabetic mice using a dose of 5 mg/kg at a frequency of six doses per week for 8 weeks[39]. Etomoxir (20 mg/kg) and C75 (15 mg/kg) were dosed (IP) three times per week for 3 weeks in GR^ECKO and control littermates.

A single IP dose of 200 mg/kg STZ was used to induce diabetes in CD-1 mice. In one experiment, male mice were randomized to one of the four groups 16 weeks after induction of diabetes: (1) untreated, (2) fenofibrate (100 mg/kg), (3) simvastatin (40 mg/kg), or (4) Wnt inhibitor (LGK974; 5 mg/kg). In each case, mice were treated for 4 weeks and compared to untreated diabetic CD-1 mice. In

another experiment, male diabetic CD-1 mice were randomized to one of the three groups: (1) untreated (vehicle), (2) etomoxir (20 mg/kg), or (3) C75 (15 mg/kg); in each case mice were treated (IP) three times/week for a total 4 weeks.

IL-6 IgG and control IgG were injected IP three times/week for a total 4 weeks at a dose of 3 mg/kg in both nondiabetic and diabetic mice. All mice had free access to food and water during experiments. Blood was obtained by retro-orbital bleed during experiments. Blood glucose was measured by glucose-strips. Urine albumin levels were assayed using a Mouse Albumin ELISA Kit (Exocell, Philadelphia, PA).

Tissues and blood were harvested at the time of sacrifice. Some kidneys were minced and stored at −80 °C for gene expression and protein analysis. Other kidneys were placed immediately in optimal cutting temperature compound for frozen sections or 4% paraformaldehyde for histologic staining.

**Bilateral adrenalectomy**. Sixteen-week-STZ-injected (24 weeks old) diabetic mice and nondiabetic CD-1 and C57BL/6 mice were used for adrenalectomy and sham operations. Animals were handled daily during the last week before experimentation to reduce stress responses. Buprenorphine was used as an analgesic. The first dose was administered 30 min before surgery and then every 12 h for 72 h, at a dose of 0.05 mg/kg subcutaneously. For the first 24 h after surgery, mice were given drinking water containing 0.9% saline to counter the effect of mineralocorticoid removal. Adrenalectomized animals were sacrificed 4 weeks after surgery. Blood was withdrawn in the morning between 9:00 and 10:00 a.m. Corticosterone levels were measured using an ELISA from Cayman Chemical (Ann Arbor MI).

**Mouse model of unilateral ureteral obstruction (UUO)**. UUO surgical procedure was performed as previously described[19]. Briefly, mice were anesthetized with isoflurane (3–5% for induction and 1–3% for maintenance). Mice were shaved on the left side of the abdomen, a vertical incision was made through the skin with a scalpel, and the skin was retracted. A second incision was made through the peritoneum to expose the kidney. The left ureter was ligated twice 15 mm below the renal pelvis with surgical silk, and the ureter was then severed between the two ligatures. Then, the ligated kidney was placed gently back into its correct anatomical position, and sterile saline was added to replenish loss of fluid. The incisions were sutured and mice were individually caged. Buprenorphine was used as an analgesic. The first dose was administered 30 min before surgery and then every 12 h for 72 h, at a dose of 0.05 mg/kg subcutaneously. Mice were sacrificed and kidney and blood samples were harvested after perfusion with phosphate-buffered saline (PBS) at 10 days after UUO. Contralateral kidneys were used as a nonfibrotic control for all experiments using this model.

**Blood pressure measurement**. Measurements were taken using the tail cuff method according to the manufacturer's instructions. Briefly, mice were trained for 5 days before measurement of blood pressure. After mice were placed in the restraint platform, which was maintained at 33–34 °C, the tail was placed through the optical sensor and the cuff compressed. The instrument automatically measured the blood pressure and repeated this ten times. Data are presented as the average of ten measurement cycles.

**Lipid analysis**. Mice were fasted for 12–15 h and blood was collected by retro-orbital venous puncture. Whole blood was spun down and plasma stored at −80 °C. Total cholesterol and triglyceride levels were measured enzymatically by kits from Wako and Sigma, respectively, according to the manufacturer's instructions.

**Morphological evaluation**. A point-counting method was utilized to evaluate the relative area of the mesangial matrix. We analyzed PAS-stained glomeruli from each mouse using a digital microscope screen grid containing 540 (27 × 20) points. Masson's trichrome-stained images were evaluated by ImageJ software, and the fibrotic areas were estimated.

**Sirius red staining**. Deparaffinized sections were incubated with picrosirius red solution for 1 h at room temperature. The slides were washed twice with acetic acid solution for 30 s per wash. The slides were then dehydrated in absolute alcohol three times, cleared in xylene, and mounted with a synthetic resin. Sirius red staining was analyzed using ImageJ software, and fibrotic areas were quantified.

**Immunohistochemistry**. Paraffin-embedded kidney sections (5 μm thick) were deparaffinized and rehydrated (2 min in xylene, four times; 1 min in 100% ethanol, twice; 1 min in 95% ethanol; 45 s in 70% ethanol; and 1 min in distilled water), and the antigen was retrieved in a 10 mM citrate buffer pH 6 at 98 °C for 60 min. To block the endogenous peroxidase, all sections were incubated in 0.3% hydrogen peroxide for 10 min. The immunohistochemistry was performed using a Vectastain ABC Kit (Vector Laboratories, Burlingame, CA). Mouse anti-β-catenin antibody (1:100) and CPT1a (Abnova; H00001374-DO1P; 1:100) antibodies were used. In the negative controls, the primary antibody was omitted and replaced with the blocking solution.

**Immunofluorescence**. Frozen kidney sections (5 μm) were used for immuno-fluorescence; double-positive labeling with CD31/αSMA (1:100/1:500), CD31/TGFβR1(1:100/1:500), and E-cadherin/αSMA (1:500/1:500) was measured. Briefly, frozen sections were dried and placed in acetone for 10 min at −30 °C. Once the sections were dried, they were washed twice in PBS for 5 min and then blocked in 2% bovine serum albumin (BSA)/PBS for 30 min at room temperature. Thereafter, the sections were incubated in primary antibody (1:100) for 1 h and washed in PBS (5 min) three times. Next, the sections were incubated with the secondary anti-bodies for 30 min, washed with PBS three times (5 min each), and mounted with mounting medium with DAPI (Vector Laboratories, Burlingame, CA). The immuno-labeled sections were analyzed by fluorescence microscopy. For each mouse, original magnification of ×400 pictures was obtained from six different areas, and quantification was performed.

**EndMT and EMT detection**. Frozen sections (5 μm) were used for the detection of EndMT and EMT. Cells undergoing EndMT were detected by double-positive labeling for CD31 (1:100) and αSMA (1:500) and/or TGFβR1 (1:500). Cells undergoing EMT were detected by double-positive labeling for E-cadherin (1:500) and αSMA (1:500). Sections were analyzed and quantified by fluorescence microscopy.

**Isolation of endothelial cells**. ECs from the kidneys of nondiabetic and diabetic mice were isolated using a standardized kit (Miltenyl Biotech, USA) by following the manufacturer's instructions. Briefly, kidneys were isolated and minced into small pieces. Using a series of enzymatic reactions by treating the tissue with trypsin and Collagenase type I solution, a single cell suspension was created. The pellet was dissolved with CD31 magnetic beads and the CD31-labeled cells were separated on a magnetic separator. The cells were further purified on a column. Cell number was counted by hemocytometer and cells were plated on 0.1% gelatin coated Petri dishes. Cell purity was measured by flow cytometry (BD FACSDiva) using PE-conjugated CD31 (BDB553373) (1:100) and FITC-conjugated CD45 (BDB553079) (1:100), both from Becton Dickinson (USA).

**Isolation of kidney TECs**. After sacrifice, kidneys from diabetic GR$^{ECKO}$ and control littermates were excised and perfused with (10 mL) followed by collagenase type II digestion (2 mg/mL). After digestion, the cortical region of kidneys was used for further processing. The cortical region of kidneys was minced and further digested in collagenase buffer for an additional 5 min at 37 °C with rotation to release cells. Digested tissue and cell suspension were passed through a 70-μm cell strainer, centrifuged at $50 \times g$ for 5 min, and washed in PBS for two rounds to collect TECs. Isolated TECs were seeded onto collagen-coated Petri dishes and cultured in renal epithelial cell medium (C-26130, PromoCell) supplemented with growth factors for TEC growth.

**Cellular bioenergetic analysis**. FAO-associated OCR was studied using extra-cellular flux analysis (Seahorse XFe96, Agilent Technologies). On the assay day, substrate-limited medium was replaced with Krebs–Henseleit buffer assay medium supplemented with 0.2% carnitine for 1 h at 37 °C without CO$_2$. Finally, just before starting the assay, BSA or 200 mM palmitate-BSA FAO substrate was added. After the assay, protein was extracted from wells with 0.1% NP-40–PBS solution and quantified with a bicinchoninic acid protein assay (Thermo Fisher Scientific) for data normalization. OCR was determined by normalizing the measurements to cell counts by quantifying the Hoechst staining in each well[64].

**ATP measurement**. ATP content was determined using the ATP Colorimetric Assay kit (Biovision), following the manufacturer's instructions.

**RNA isolation and qPCR**. Total RNA was isolated using standard Trizol protocol. RNA was reverse transcribed using the iScript cDNA Synthesis kit (Bio-Rad) and qPCR was performed on a Bio-Rad C1000 Touch thermal cycler using the resultant cDNA, as well as qPCR Master mix and gene-specific primers. The list of mouse primers used is in Table S1.

Results were quantified using the delta–delta-cycle threshold (Ct) method (ΔΔCt). All experiments were performed in triplicate and 18S was utilized as an internal control.

**Western blot**. Protein lysates were boiled in sodium dodecyl sulfate (SDS) sample buffer at 94 °C for 5 min. After centrifugation at $17,000 \times g$ for 10 min at 4 °C, the supernatant was separated on 6–12% SDS polyacrylamide gels, and blotted onto PVDF membranes (Immobilon, Bedford, MA) via the semidry method. After blocking with TBS (Tris buffered saline containing 0.05% Tween 20) containing 5% BSA, membranes were incubated with each primary antibody (GR: 1:1000; Anti-TGFβR1: 1:500; anti-αSMA: 1:500; anti-vimentin: 1:2000, anti-β-catenin: 1:500, anti-β-actin 1:10,000), in TBS containing 5% BSA at 4 °C overnight. Protein bands were visualized using the Odyssey Infrared Imaging System (LI-COR Biotechnol-ogy), and enhanced chemiluminescence detection system (Pierce Biotechnology, Rockland IL) using ImageQuant LAS 400 (GE Healthcare Life Sciences, Uppsala, Sweden). Densitometry was performed using ImageJ software (NIH).

**In vitro experiments and siRNA transfection**. HUVECs were used at passages four to eight and cultured in Endothelial Basal Medium-2 media with growth factors and 10% serum. Human GR-specific siRNA (Invitrogen) was used at a concentration of 100 nM for 48 h to effectively knockdown GR. Cells were treated with or without TGFβ2 (10 ng/ml) for 48 h and harvested for western blot analysis. Some transfected cells were treated with fenofibrate (1 μM) and etomoxir (40 μM) for 48 h. In a second set of experiments, Human HK-2 cells were cultured in DMEM and Keratinocyte-SFM (1X) medium (Life Technologies Green Island NY). When the cells reached 70% confluence, CM from control siRNA and GR siRNA-transfected HUVECs was added to the HK-2 cell culture.

**Fatty acid uptake**. Cultured isolated kidney ECs were incubated with medium containing 2 μCi [$^{14}$C]-palmitate. [$^{14}$C]-palmitate uptake was measured by liquid scintillation counting.

**Fatty acid oxidation**. Cultured isolated kidney ECs were incubated with medium containing 0.75 mmol/L palmitate (conjugated to 2% free FA–free BSA/[$^{14}$C] palmitate at 2 μCi/mL) for 2 h. 1 mL of the culture medium was transferred to a sealable tube, the cap of which housed a Whatman filter paper disc. $^{14}$CO$_2$ trapped in the media was then released by acidification of media using 60% perchloric acid. Radioactivity that had become adsorbed onto the filter discs was then quantified by liquid scintillation counting.

**Statistical analysis**. All values are expressed as means ± SEM and analyzed using the statistical package for the GraphPad Prism 7 (GraphPad Software, Inc., La Jolla, CA). One-way ANOVA, followed by Tukey's test and two-way ANOVA (Figs. 1 and 2), was employed to analyze significance when comparing multiple independent groups. The post hoc tests were run only if $F$ achieved $p < 0.05$ and there was no significant variance in homogeneity. In each experiment, $N$ represents the number of separate experiments (in vitro) and the number of mice (in vivo). Technical replicates were used to ensure the reliability of single values. Data analyses were blinded. The data were considered statistically significant at $p < 0.05$.

**Reporting summary**. Further information on research design is available in the Nature Research Reporting Summary linked to this article.

## Data availability
Source data are provided with this paper. All relevant data are available from the authors upon reasonable request.

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

## Acknowledgements

This work is supported by grants from the National Institutes of Health to A.D. (R01HL128406), C.F.-H. (R35HL135820), and J.G. (R01HL131952).

## Author contributions

S.P.S. performed experiments and wrote the paper. H.Z. helped in genotyping the mice and was involved in validation of the data. O.S. performed surgical experiments in mice. B.L. performed cell isolation experiments. K.K., D.K., A.D., and C.F.-H. provided intellectual input. J.G. mentored, validated the data, made intellectual contributions, wrote the paper, and performed final editing of the paper.

## Competing interests

The authors declare no competing interests.
