## [Peer Review File · Nature Communications]

Reviewers' comments:

Reviewer #1 (Remarks to the Author):

This study examined the protective role of glucocorticoid receptor (GR) against renal fibrosis by inhibiting renal inflammation, EndoMT, and EMT in diabetic nephropathy models and the UUO model. Studies were also performed in different wild-type mouse strains. Overall, the studies demonstrated that diabetic GR knock-out mice showed overactivation of the Wnt pathway, inflammation, and metabolism disorders. However, the mechanistic links for the activation of the Wnt pathway and fibrosis in diabetes or UUO have not been established.

1. To study the importance of endothelium-derived GR in renal fibrosis, C57BL/6 mice and CD-1 mice were compared. Diabetic CD-1 mice develop severe renal fibrosis compared with that in diabetic C57 mice. However, the mechanism is unknown. Only the difference in GR expression between the two mouse strains is not sufficient to support the conclusion that endothelial GR deficiency results in the renal fibrogenic phenotype in diabetic mice. GOF and LOF experiments are required.
2. In the kidney, there are two capillary systems, the glomerular capillary bed, and peritubular capillaries. Please specify which endothelium-derived GR is most important in diabetic renal fibrosis at least in Discussion.
3. The effect of Wnt pathway activation in diabetic nephropathy is controversial. In glomeruli, previous studies showed that Wnt pathway activation conferred protective effects in the diabetic kidney. However, other studies demonstrated that aberrant Wnt signaling activation in the tube results in renal fibrosis. In this study, the authors studied the Wnt activation in tubular epithelial cells. As GR was deleted in all kidney vascular endothelial cells, the change of glomerular Wnt activation in diabetic mice after GR deletion needs to be demonstrated.
4. The data on fatty acid uptake and FAO are interesting. In this study, fatty acid uptake was increased in diabetic EC, and FAO was decreased. As the receptor of Fenofibrate, PPAR α regulates FAO-related gene transcription. CPT1 is one of the PPAR α -target genes and works on fatty acid transportation. It is reported that PPAR α was decreased in diabetic kidneys. In supple Figure 5f, the expression of CPT1 was decreased in the diabetic kidneys, which was prevented by the Fenofibrate treatment. The in vivo data suggested that fatty acid uptake should be decreased in diabetic kidneys and fenofibrate could increase it back. Then the conclusion from Figure S5 would be the opposite of the in vitro data on primary cultured endothelial cells in Figure 2. Why primary cultured cells showed different results than the in vivo data? To answer the question, endothelial CPT1 should be measured.
5. In the manuscript, experimental groups in data presentation should be consistent. Please present the data on non-diabetic groups with the diabetic groups.
6. Although mouse renal vascular ECs were cultured by CD31 beads, the cells still could be contaminated with macrophage, fibroblast, pericytes, etc. Cell purity should be examined and presented.
7. The histological images should be consistently captured and presented either in the renal cortex or in the medulla. Please check the images in Figure 2j, 6d, 6e, supple Figure 3, s5a and 5f.
8. Figure 2j, the representative MTS image of diabetic GR KO mice didn't match with the bar graph. There is no difference between the diabetic control mice and diabetic KO group.
9. Figure 2k and i are small and of poor quality. Please describe what the image in the white frame presented for.
10. Figure 4b, why the Western blot analysis data showed two blotting bands for β -catenin? The total β -catenin level is not an ideal marker for Wnt pathway activation. β -catenin nuclear translocation or non-phosphorylated-catenin should be examined.
11. Figure 4c, there are almost no CD31+ cells in the representative diabetic control image for TGF β R1. Please check the raw data and replace the image.
12. Figure 5c and Figure 6g are too small and not in good quality. Statistical analysis should be performed and presented.
13. Figure 7c. E cadherin is a membrane protein and SMA locates in the cytosol. The staining showed both proteins colocalized with DAPI. The images should be replaced.
14. The labels of the y axis were missing in Figure 7b, g, i.
15. Supplemental figure 3g, h, 5f presented the β -catenin expression by immunostaining. Normally, β -catenin is colocalized with E-cadherin and expressed in the cell membrane. Why there is no β -catenin expression signal in the normal kidney.
16. The scale bar in Figure 2, 4, 5, 6. S2, S3, should not be 50 mm. Please correct it.

Reviewer #2 (Remarks to the Author):

The manuscript examined the role of endothelial glucocorticoid receptor in diabetic kidney disease development.

DKD is responsible for close half of CKD and ESRD cases and therefore the topic is highly significant and relevant.

The team did a really good job generating mice with endothelial specific GR deletion and making these animals diabetic.

The team had used the diabetic ApoE mice model and performed a careful phenotype analysis showing increased disease severity in KO diabetic mice. These are very nicely performed and presented studies.

The team has performed a very detailed analysis by looking into variety of cytokine levels.

Figure3 is extremely busy and poorly organized. I think those cytokines with significant changes in the diabetic DKO should be shown. Cytokines should be organized by groups.

The data with the Wnt pathway is not connected to the earlier data. It is unclear why the authors had focused on Wnt.

GR is a transcription factor with ample of CHIPseq showing its direct transcriptional targets. How do the direct GR targets change in diabetic mice?

The Wnt inhibitor experiment is really nice, but again leaves us a bit uncertain about the mechanism or the target. These global Wnt inhibition studies, so multiple cell types could be targeted.

The fatty acid oxidation studies are interesting but again hard to connect to direct GR targets, most FAO occurs in the tubules.

The authors also missed the opportunity to examine some obvious endothelial phenotypes such as endothelial density, sprouting, leakiness, any change in oxygenation etc.

Was there as change in blood pressure in these animals.

Most importantly if the data is correct one would predict that glucocorticoids would improve diabetic kidney disease. Do they?

I know that glucocorticoids make diabetes worse, but maybe they can be tested for other kidney disease?

Finally, I would like to know whether any of these finding can be recapitulated or replicated in patients samples.

Reviewer #3 (Remarks to the Author):

All throughout the paper, small n-values are used. I think it is risky to draw conclusions based on experiments performed with 5-6 mice per group, and moreover not applying the correct (2-way ANOVA) statistical test.

In the text accompanying Figure 1, it is written that 'CD-31 positive cells from...'. This is not really correct. It should be rephrased. There is less GR staining in CD-1 mice in DM conditions compared to Control conditions.

The data in Figure 1 should not be studied by One-way ANOVA but by two-way ANOVA. There are two conditions and two types of mice. The statistics must be reconsidered.

The Western blot in Figure 1 is misleading. It probably represents the samples with most

outspoken effect. A Western blot must be shown with all the samples. This blot is available as it was used to generate the bar plot.

The text accompanying Figure 2 is not clear, at least the sentence 'Diabetes was induced...' must be revised, because there are some mistakes when describing the mouse groups.

In Figure 2, like in Figure 1, always 2-way ANOVA must be applied.

Since the paper starts by showing that there is a spectacular difference between C57BL/6J and CD-1 mice in terms of development of DM-associated fibrosis, and that this may be related to GR disappearance, it is imperative that the authors mention at least something about the genetic background of their GR and other mutant mice.

In Figure 3, a great number of cytokine measurements are shown, in serum of CD-1 mice as well as C57BL/6J mice, both controls and diabetic mice. The added value of all these measurements is unclear since the authors do not consider more than a couple of cytokines. Pro-inflammatory cytokines such as IL1, IL6, IL17, TNF are all rising in CD-1 but are decreasing in C57BL/6J. To me, this suggests that in both mouse lines (CD-1 is not an inbred line, which further complicates this study), the HPA axis is activated, leading, in C57BL/6J to an anti-inflammatory response. Since GR becomes greatly depleted in CD-1 mice, the cortisol which is generated may have no anti-inflammatory and protective effect. This looks very conceivable and should be studied. The authors should measure corticosterone in blood in the 4 groups of mice, rather than all these cytokines and they should perform adrenalectomy in C57BL/6J and CD-1 mice, and then study if the difference in response to streptozotocin disappears.

In the second part of Figure 3, the mutant mice are then studied in terms of cytokines, but the control levels are lacking (mice not treated with streptozotocin, all 4 groups). These can not simply be deduced from Figure 3a. These controls are essential and must really be added.

The authors measure the expression of Wnt target genes by qPCR, protein and IHC in Figure 4, and find that GR-KO mice and double mutant mice have increased expression of these genes/proteins. Once again, the unstimulated controls are lacking and the data hence can not be interpreted. Another obvious question that arises while reading this part of the paper is how these Wnt markers are expressed in control and diabetes samples from the CD-1 and C57BL/6J mice.

In Figure 5, the impact of a Wnt inhibitor is studied in kidney weight and pathology in GR^{fl/fl} and GRECKO mice treated with streptozotocin. Once again, untreated controls are lacking. The drug has some protective effect. To me, it seems hardly believable that statistical significance is obtained, with such small groups and small effects. Also, since the drug has protective effects in GR^{fl/fl} mice, and these are probably on a C57BL/6J background, considered to be a resistant background, what are we looking at here? This requires further explanation and investigation. The effect of genetic GR depletion seems clear in the paper, but the effect of LGK974 appears to be independent of GR presence or absence.

The final part of the paper studies metabolic pathways and PPAR α dependent genes in GR knock-down cells. The authors should study all this in endothelial cells derived from GR^{fl/flo} and GRECKO mice.

Reviewer #1 (Remarks to the Author):

This study examined the protective role of glucocorticoid receptor (GR) against renal fibrosis by inhibiting renal inflammation, EndoMT, and EMT in diabetic nephropathy models and the UUO model. Studies were also performed in different wild-type mouse strains. Overall, the studies demonstrated that diabetic GR knock-out mice showed overactivation of the Wnt pathway, inflammation, and metabolism disorders. However, the mechanistic links for the activation of the Wnt pathway and fibrosis in diabetes or UUO have not been established.

1. To study the importance of endothelium-derived GR in renal fibrosis, C57BL/6 mice and CD-1 mice were compared. Diabetic CD-1 mice develop severe renal fibrosis compared with that in diabetic C57 mice. However, the mechanism is unknown. Only the difference in GR expression between the two mouse strains is not sufficient to support the conclusion that endothelial GR deficiency results in the renal fibrogenic phenotype in diabetic mice. GOF and LOF experiments are required.

We thank the reviewer for these comments. Kidneys of diabetic CD-1 mice show increased fibrosis when compared to the kidneys of diabetic C57BL/6 mice which was associated with increased endothelial-mesenchymal transition and altered pro-inflammatory responses (Fig S10). We tried to investigate this strain-specific difference further. We observed that IL-6 was a key mesenchymal inducer which was associated with fibrogenic processes. Diabetic CD-1 had elevated IL-6 levels when compared to nondiabetic CD-1 control mice and diabetic C57BL/6 mice. Elevation of IL-6 was also observed in diabetic GR^{ECKO}, diabetic DKO mice compared to their diabetic littermate controls. IL-6 neutralization in diabetic GR^{ECKO}, diabetic DKO mice and diabetic CD-1 mice significantly ameliorated the renal fibrosis, suggesting that IL-6 is key pro-inflammatory cytokines which is released upon loss of GR in the endothelium (Fig 3; Fig S12 and Fig S13).

In addition, we performed bilateral adrenalectomy to analyze the effects of corticosterone suppression on the progression of renal fibrosis. While the adrenalectomy procedure was successful, as evidenced by suppression of corticosterone, it did not cause any effect on the fibrosis level in any of the mice studied suggesting that the tissue-specific effects of GR in the endothelium superseded the global suppression of corticosterone. These data are presented in Figure S1.

To study the gain of function of GR in endothelial cells, we overexpressed GR in cultured HUVECs. GR overexpression significantly suppressed the TGFβ2 induced fibrogenic process and associated higher Wnt signaling, suggesting that GR overexpression has a protective and anti-mesenchymal role in endothelial cells (Fig S28).

To study the loss of function of GR further, we extended our studies to include

GR^{ECKO} mice on the Apoe^{-/-} background (DKO mice). The kidneys of diabetic GR^{ECKO} and diabetic DKO mice have profound fibrosis compared to endothelial GR-replete controls. These data are presented in Figure 2.

2. In the kidney, there are two capillary systems, the glomerular capillary bed, and peritubular capillaries. Please specify which endothelium-derived GR is most important in diabetic renal fibrosis at least in Discussion.

We thank the reviewer for the comment. We analyzed the glomerular capillary bed and peritubular capillaries. The loss of GR in the endothelium enhanced both glomerular fibrosis and interstitial fibrosis (**Fig 2; Fig S5 and Fig S14b**). However, our focus was more on studying the role of peritubular capillaries in the regulation of interstitial fibrosis in the kidneys. We have described the contribution of both the glomerular capillary bed and peritubular capillaries in the diabetic renal fibrosis.

3. The effect of Wnt pathway activation in diabetic nephropathy is controversial. In glomeruli, previous studies showed that Wnt pathway activation conferred protective effects in the diabetic kidney. However, other studies demonstrated that aberrant Wnt signaling activation in the tubule results in renal fibrosis. In this study, the authors studied the Wnt activation in tubular epithelial cells. As GR was deleted in all kidney vascular endothelial cells, the change of glomerular Wnt activation in diabetic mice after GR deletion needs to be demonstrated.

We thank the reviewer for these comments and agree that there is a controversial role of Wnt signaling in diabetic nephropathy. Our data demonstrate that Wnt signaling is aberrant both in isolated peritubular endothelial cells and glomerular endothelial cells. We performed PAS staining and immunohistochemical analysis of vimentin and found features of segmental glomerular fibrosis and interstitial fibrosis in diabetic kidneys from GR^{ECKO} mice. This segmental glomerular fibrosis was accompanied by a higher level of Wnt signaling in the glomeruli. Isolated endothelial cells from the glomeruli and interstitium of diabetic GR^{ECKO} kidneys have higher Wnt signaling when compared to diabetic control mice. These data have been added in the revised manuscript in Figures S5 and S14 and Figure 4.

4. The data on fatty acid uptake and FAO are interesting. In this study, fatty acid uptake was increased in diabetic EC, and FAO was decreased. As the receptor of Fenofibrate, PPAR α regulates FAO-related gene transcription. CPT1 is one of the PPAR α -target genes and works on fatty acid transportation. It is reported that PPAR α was decreased in diabetic kidneys. In supple Figure5f, the expression of CPT1 was decreased in the diabetic kidneys, which was prevented by the Fenofibrate treatment. The in vivo data suggested that fatty acid uptake should be decreased in diabetic kidneys and fenofibrate could increase it back. Then the conclusion from Figure S5 would be the opposite of the in vitro data on primary cultured endothelial cells in Figure2. Why

primary cultured cells showed different results than the in vivo data? To answer the question, endothelial CPT1 should be measured.

We thank the reviewer for this comment. We analyzed fatty acid uptake in isolated endothelial cells and observed that fatty acid uptake was higher in the isolated endothelial cells from kidneys of diabetic GR^{ECKO}. This higher intracellular level of fatty acid might cause disruption of endothelial cell metabolism. Most fatty acids are transported into kidney endothelial cells by FATP1, FATP4 and CD36. We found no significant difference in the expression of FATP1, FATP4 or CD36 in isolated cells from diabetic GR^{ECKO} when compared to diabetic control littermates (Figure S24).

We measured CPT1a, which was unaltered in nondiabetic endothelial cells but suppressed in endothelial cells from the kidneys of diabetic GR^{ECKO} when compared to the diabetic control littermates. These data suggest higher fatty acid uptake in these GR KO cells. These data are shown in Figure 4a and Figure 7i.

5. In the manuscript, experimental groups in data presentation should be consistent. Please present the data on non-diabetic groups with the diabetic groups.

We agree with the reviewer and are sorry for this oversight. We have corrected this in all applicable figures.

6. Although mouse renal vascular ECs were cultured by CD31 beads, the cells still could be contaminated with macrophage, fibroblast, pericytes, etc. Cell purity should be examined and presented.

We performed FACS analysis to quantify the purity of our endothelial cell isolation. These data are presented in Figure S2.

7. The histological images should be consistently captured and presented either in the renal cortex or in the medulla. Please check the images in Figure 2j, 6d, 6e, supple Figure 3, s5a and 5f.

We agree with the reviewer. We have verified that all representative images are now from the renal cortex.

8. Figure 2j, the representative MTS image of diabetic GR KO mice didn't match with the bar graph. There is no difference between the diabetic control mice and diabetic KO group.

We have corrected and replaced the representative images that now match the bar graph.

9. Figure 2k and i are small and of poor quality. Please describe what the image in the white frame presented for.

We performed additional staining and have replaced these images with newer ones of better quality now found in Figure S7.

10. Figure 4b, why the Western blot analysis data showed two blotting bands for β -catenin? The total β -catenin level is not an ideal marker for Wnt pathway activation. β -catenin nuclear translocation or non-phosphorylated-catenin should be examined.

We have performed additional Western blotting experiments and now present both total β -catenin and non-phosphorylated (active) β -catenin. We have added the results in the revised manuscript (Figure 4a-b, Figure 7b,i).

11. Figure 4c, there are almost no CD31+ cells in the representative diabetic control image for TGF β R1. Please check the raw data and replace the image.

We have replaced these images as suggested (Figure 4c).

12. Figure 5c and Figure 6d are too small and not in good quality. Statistical analysis should be performed and presented.

As per the reviewer's suggestion, we have replaced the images with higher resolution images and have added the quantification.

13. Figure 7c. E cadherin is a membrane protein and SMA locates in the cytosol. The staining showed both proteins colocalized with DAPI. The images should be replaced.

We have improved the quality of the images which now clearly shows that neither protein co-localizes with DAPI (Figure S22).

14. The labels of the y axis were missing in Figure 7b, g,i.

We are sorry for inconvenience. We have corrected it.

15. Supplemental figure 3g, h, 5f presented the β -catenin expression by immunostaining. Normally, β -catenin is colocalized with E-cadherin and expressed in the cell membrane. Why there is no β -catenin expression signal in the normal kidney.

We analyzed activated (nuclear) β -catenin in Supplemental figure 3g, h, 5f (now S16g,h and S25f). Standardization of the activated β -catenin IHC method resulted in very low signal in normal kidneys.

16. The scale bar in Figure 2,4,5, 6.S2, S3, should not be 50 mm. Please correct it.

We apologize for the inconvenience and have corrected it throughout the manuscript.

Reviewer #2 (Remarks to the Author):

The manuscript examined the role of endothelial glucocorticoid receptor in diabetic kidney disease development.

DKD is responsible for close half of CKD and ESRD cases and therefore the topic is highly significant and relevant.

The team did a really good job generating mice with endothelial specific GR deletion and making these animals diabetic.

The team had used the diabetic ApoE mice model and performed a careful phenotype analysis showing increased disease severity in KO diabetic mice. These are very nicely performed and presented studies.

The team has performed a very detailed analysis by looking into variety of cytokine levels.

Figure3 is extremely busy and poorly organized. I think those cytokines with significant changes in the diabetic DKO should be shown. Cytokines should be organized by groups.

We thank the reviewer for the suggestion. We have re-organized Figure 3 into Figures S10, S11 and S17. We focused further efforts on the key cytokine of IL-6 and performed IL-6 neutralization experiments in diabetic GR^{ECKO} and diabetic DKO mice. IL-6 neutralization completely abolished the fibrogenic processes in the diabetic kidneys. These data are now shown in Figure 3 and Figures S12 and S13 provide additional supporting data.

The data with the Wnt pathway is not connected to the earlier data. It is unclear why the authors had focused on Wnt.

The rationale for focusing on Wnt signaling was derived from a recent publication from the lab (PMC7098785/Ref #25) in which we demonstrated by next-gen sequencing that loss of endothelial GR upregulates canonical Wnt signaling. As current literature supports a role for Wnt signaling in the development of renal fibrosis, we aimed to study whether GR-Wnt interactions could influence this phenotype. We haven integrated this data more logically into our work in Figures 4 and S14.

GR is a transcription factor with ample of CHIPseq showing its direct transcriptional targets. How do the direct GR targets change in diabetic mice?

We identified that loss of endothelial GR caused increased mRNA expression of transcription factor Snail1, and other Snail1-associated fibrogenic genes such as TGFB1, aSMA and HIF1 α . We have added this information in the revised manuscript in Figures S6, S14, S20 and S21.

The Wnt inhibitor experiment is really nice, but again leaves us a bit uncertain about the mechanism or the target. These global Wnt inhibition studies, so multiple cell types

could be targeted.

We agree with this comment. In this study, we have shown that the Wnt inhibitor completely abolished both EndMT and EMT in the kidneys of diabetic control mice. However, the Wnt inhibitor abolished only EMT, not EndMT, in the kidneys of diabetic GR^{ECKO} suggesting that endothelial cells are a key cell type in the progression of renal fibrosis. We are developing an endothelial-specific LRP5/6 (canonical Wnt) KO mouse which will be the subject of a future study and using this model we will be able to more precisely identify the specific role of endothelial cells in this phenomenon.

The fatty acid oxidation studies are interesting but again hard to connect to direct GR targets, most FAO occurs in the tubules.

Our data demonstrate loss of GR causes significant downregulation in PPAR α and PPAR α -dependent genes and therefore, significant suppression in the level of fatty acid oxidation in endothelial cells. GR loss-linked suppression of fatty acid oxidation contributes to EndMT in endothelial cells. This GR loss-linked EndMT causes activation of EMT events with significant suppression of FAO in tubular epithelial cells which influences fibrosis in whole kidneys. These data have been added in the revised manuscript (Fig 4a and Fig 6, Fig 7b and 7i; Fig S6 and Fig S14). Our data also show that the PPAR α agonist fenofibrate ameliorated renal fibrosis in our model by elevating the endothelial GR level (Fig S25a and Fig S25b).

The authors also missed the opportunity to examine some obvious endothelial phenotypes such as endothelial density, sprouting, leakiness, any change in oxygenation etc.

Was there as change in blood pressure in these animals.

We thank the reviewer for these suggestions. Endothelial density and permeability studies were performed and are presented in Figure S15. Blood pressure was also measured and shown in Figure S4.

Most importantly if the data is correct one would predict that glucocorticoids would improve diabetic kidney disease. Do they?

I know that glucocorticoids make diabetes worse, but maybe they can be tested for other kidney disease?

Yes, we suspect if glucocorticoids could be administered in an endothelial-specific manner that diabetes would be improved. We attempted to treat diabetic CD-1 mice with dexamethasone but observed significant mortality in these animals. We were able to treat mice subjected to UUO with dexamethasone and observed significant improvement in the degree of fibrosis in these animals. This data is presented in Figure S9.

Finally, I would like to know whether any of these finding can be recapitulated or replicated in patients samples.

At this time we are unable to access patient samples, but these studies will be pursued in the future.

Reviewer #3 (Remarks to the Author):

All throughout the paper, small n-values are used. I think it is risky to draw conclusions based on experiments performed with 5-6 mice per group, and moreover not applying the correct (2-way ANOVA) statistical test.

We have increased the mouse number to 8 in nearly all of the key experiments. Statistics have been reviewed and we have analyzed the data with two-way ANOVA wherever we used two types of mice and two different time points.

In the text accompanying Figure 1, it is written that 'CD-31 positive cells from...'. This is not really correct. It should be rephrased. There is less GR staining in CD-1 mice in DM conditions compared to Control conditions.

We thank the reviewer for this comment. In the kidneys of diabetic CD-1 mice, the GR protein level was significantly suppressed in tubules and endothelium as compared to nondiabetic controls. We corrected the label in Figure 1 (Y axis).

The data in Figure 1 should not be studied by One-way ANOVA but by two-way ANOVA. There are two conditions and two types of mice. The statistics must be reconsidered.

These statistics have been revised.

The Western blot in Figure 1 is misleading. It probably represents the samples with most outspoken effect. A Western blot must be shown with all the samples. This blot is available as it was used to generate the bar plot.

We have added the correct blot in Figure 1.

The text accompanying Figure 2 is not clear, at least the sentence 'Diabetes was induced...' must be revised, because there are some mistakes when describing the mouse groups.

We have corrected the mouse groups and clarified the language.

In Figure 2, like in Figure 1, always 2-way ANOVA must be applied.
These statistics have been revised as recommended.

Since the paper starts by showing that there is a spectacular difference between

C57BL/6J and CD-1 mice in terms of development of DM-associated fibrosis, and that this may be related to GR disappearance, it is imperative that the authors mention at least something about the genetic background of their GR and other mutant mice.

All mouse models used in these studies, except the CD-1 mice, are on a C57BL/6 background. This detail has been clarified in the manuscript.

In Figure 3, a great number of cytokine measurements are shown, in serum of CD-1 mice as well as C57BL/6J mice, both controls and diabetic mice. The added value of all these measurements is unclear since the authors do not consider more than a couple of cytokines. Pro-inflammatory cytokines such as IL1, IL6, IL17, TNF are all rising in CD-1 but are decreasing in C57BL/6J. To me, this suggests that in both mouse lines (CD-1 is not an inbred line, which further complicates this study), the HPA axis is activated, leading, in C57BL/6J to an anti-inflammatory response. Since GR becomes greatly depleted in CD-1 mice, the cortisol which is generated may have no anti-inflammatory and protective effect. This looks very conceivable and should be studied. The authors should measure corticosterone in blood in the 4 groups of mice, rather than all these cytokines and they should perform adrenalectomy in C57BL/6J and CD-1 mice, and then study if the difference in response to streptozotocin disappears.

We thank the reviewer for this comment. We simplified and arranged the cytokine panels (Fig S10 and Fig S11). We tried to investigate the pronounced difference in the fibrosis level between these two diabetic mouse strains. We observed that IL-6 was a key cytokine which was associated with fibrogenic processes in diabetes. Diabetic CD-1 mice had elevated levels of IL-6 when compared to nondiabetic CD-1 control mice and to diabetic C57BL/6 mice. IL-6 elevation was also observed in diabetic GR^{ECKO} and diabetic DKO mice when compared to their diabetic littermate controls. In addition, we performed IL-6 neutralization in GR^{ECKO}, DKO mice and CD-1 mice. IL-6 neutralization significantly ameliorated the renal fibrosis in diabetic mice, suggesting that IL-6 is a key proinflammatory cytokine which is released upon loss of GR in the endothelium. (Fig 3; Fig S12 and Fig S13).

In addition, we performed bilateral adrenalectomy to analyze the effects of corticosterone suppression on the progression of renal fibrosis. While the adrenalectomy procedure was successful, as evidenced by suppression of corticosterone, it did not cause any effect on the fibrosis level in any of the mice studied suggesting that the tissue-specific effects of GR in the endothelium superseded the global suppression of corticosterone. These data are presented in Figure S1. In the future, we plan to study the effect of adrenalectomy in a mouse model of type II diabetes.

In the second part of Figure 3, the mutant mice are then studied in terms of cytokines, but the control levels are lacking (mice not treated with streptozotocin, all 4 groups). These cannot simply be deduced from Figure 3a. These controls are essential and must really be added.

We thank the reviewer for this suggestion. We have added these data in Fig S10 and Fig S11.

The authors measure the expression of Wnt target genes by qPCR, protein and IHC in Figure 4, and find that GR-KO mice and double mutant mice have increased expression of these genes/proteins. Once again, the unstimulated controls are lacking and the data hence cannot be interpreted. Another obvious question that arises while reading this part of the paper is how these Wnt markers are expressed in control and diabetes samples from the CD-1 and C57BL/6J mice.

We thank the reviewer for this suggestion. We have added unstimulated controls in Figures S14a-c and Figures 4a,c. We also investigated protein and mRNA expression of Wnt-dependent genes in the kidneys. Diabetic CD-1 mouse kidneys displayed significantly higher expression of these genes compared to diabetic C57BL/6 mouse kidneys. These data are presented in Figure S16a-b.

In Figure 5, the impact of a Wnt inhibitor is studied in kidney weight and pathology in GR^{fl/fl} and GRECKO mice treated with streptozotocin. Once again, untreated controls are lacking. The drug has some protective effect. To me, it seems hardly believable that statistical significance is obtained, with such small groups and small effects. Also, since the drug has protective effects in GR^{fl/fl} mice, and these are probably on a C57BL/6J background, considered to be a resistant background, what are we looking at here?

We thank the reviewer for the comment. As per reviewer's suggestion we performed Wnt inhibitor treatment in control mice and increased the number of mice. These data are presented in Figures S18 and S19.

We agree with the reviewer that it is difficult to see the protective effect of the Wnt inhibitor in GR^{fl/fl} mice. Therefore, we have performed this study in both diabetic CD-1 and the UUO model of fibrosis. mice In both of these models, the Wnt inhibitor significantly improved fibrosis. These data have been added in Figure S16c-i.

This requires further explanation and investigation. The effect of genetic GR depletion seems clear in the paper, but the effect of LGK974 appears to be independent of GR presence or absence.

We thank the reviewer for pointing this out. Our mice lack GR in endothelial cells but not in other cell types such as tubules, pericytes, podocytes etc.

The fibrotic phenotype in the kidneys of GR^{ECKO} is primarily due to endothelial-to-mesenchymal transition (EndMT) which we propose is due to 3 key mechanisms: 1) direct effects of IL-6, 2) augmented Wnt signaling and 3) due to suppressed fatty acid oxidation in endothelial cells. These cumulative effects lead to higher EndMT and hence higher fibrosis in diabetes.

In control mice (GR replete mice), Wnt inhibitor (LGK974) abolished fibrosis by inhibiting EndMT and EMT (mesenchymal program in tubules) however, in GR^{ECKO} mice (which have GR depletion only in endothelial cells) EMT events were suppressed in tubules however, EndMT was not inhibited in the endothelium. These data suggest that GR is a critical anti-EndMT molecule which is required for endothelial cell homeostasis and its loss contributes to EndMT activation. Therefore, LGK974 suppressed fibrosis in tubules but failed to suppress it in EndMT-derived fibrosis in the kidneys of diabetic GR^{ECKO} mice. These data have been added in Fig 5 and Fig S22.

The final part of the paper studies metabolic pathways and PPAR α dependent genes in GR knock-down cells. The authors should study all this in endothelial cells derived from GR^{f/f} and GRECKO mice.

We thank the reviewer for the suggestion. We have added the protein and gene expression data from PPAR α dependent genes in isolated the endothelial cells from GR^{f/f} and GR^{ECKO} mice. These data are presented in Figures 4a, 7i-j and S14.

REVIEWERS' COMMENTS

Reviewer #1 (Remarks to the Author):

The authors have responded to the comments and revised the manuscript.

Reviewer #2 (Remarks to the Author):

The authors have responded to my comments.
Nicely performed studies. Congratulations

Katalin Susztak

Reviewer #3 (Remarks to the Author):

The authors have done a good revision and replied to my concerns.

REVIEWERS' COMMENTS

Reviewer #1 (Remarks to the Author):

The authors have responded to the comments and revised the manuscript.

Reviewer #2 (Remarks to the Author):

The authors have responded to my comments.
Nicely performed studies. Congratulations

Katalin Susztak

Reviewer #3 (Remarks to the Author):

The authors have done a good revision and replied to my concerns.

Thank you. No further revisions are necessary based on these comments.